# ConceptScope: Characterizing Dataset Bias via Disentangled Visual Concepts

**Jinho Choi**
KAIST AI

**Hyesu Lim**
KAIST AI

**Steffen Schneider**[1,2]
Helmholtz Munich

**Jaegul Choo**[*]
KAIST AI

## Abstract

Dataset bias, where data points are skewed to certain concepts, is ubiquitous in machine learning datasets. Yet, systematically identifying these biases is challenging without costly, fine-grained attribute annotations. We present **ConceptScope**, a scalable and automated framework for analyzing visual datasets by discovering and quantifying human-interpretable concepts using Sparse Autoencoders trained on representations from vision foundation models. ConceptScope categorizes concepts into *target*, *context*, and *bias* types based on their semantic relevance and statistical correlation to class labels, enabling class-level dataset characterization, bias identification, and robustness evaluation through concept-based subgrouping. We validate that ConceptScope captures a wide range of visual concepts, including objects, textures, backgrounds, facial attributes, emotions, and actions, through comparisons with annotated datasets. Furthermore, we show that concept activations produce spatial attributions that align with semantically meaningful image regions. ConceptScope reliably detects known biases (e.g., background bias in Waterbirds [59]) and uncovers previously unannotated ones (e.g, co-occurring objects in ImageNet [12]), offering a practical tool for dataset auditing and model diagnostics. The code is available at https://github.com/jjho-choi/ConceptScope.

## 1   Introduction

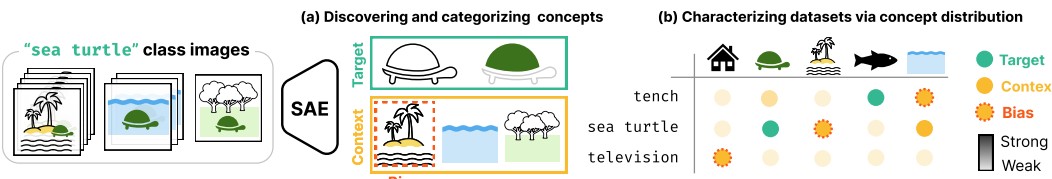

Figure 1: **Overview**. Visual concepts are unevenly distributed in image datasets, e.g., many images labeled as `sea turtle` are taken at the beach. **(a)** Our framework **ConceptScope** discovers and categorizes presenting visual concepts into *target*, *context*, and *bias* concepts based on a Sparse Autoencoder (SAE)-based concept dictionary. **(b)** We characterize datasets by averaging activation of each concept across the training set, where brighter elements indicate stronger concept presence.

Large-scale datasets have significantly driven recent breakthroughs in artificial intelligence, particularly in large language models and vision-language foundation models, as described by scaling laws [26]. Alongside increasing dataset sizes, recent studies have highlighted the importance of careful dataset curation. For instance, curated subsets of data have demonstrated favorable performance compared to larger, less carefully constructed datasets [20]. However, there is currently no

---

[*]Correspondence: jchoo@kaist.ac.kr; [1]Institute of Computational Biology, Computational Health Center, Helmholtz Munich; [2]Munich Center for Machine Learning (MCML)

39th Conference on Neural Information Processing Systems (NeurIPS 2025).

systematic framework for designing effective datasets. In practice, researchers must train models on multiple curated versions of a dataset using a fixed training protocol and compare their performances, making the process both resource-intensive and indirect.

One critical aspect influencing dataset quality is the diversity and distribution of visual concepts, defined as recurring patterns such as colors, objects, and textures. Since datasets are collected and filtered according to specific human criteria, they inevitably encode inherent biases. For example, as shown in Fig. 1, approximately 75% of images labeled as `leatherback sea turtle` in ImageNet [12] are captured on beaches, while only 15% are taken underwater. Models trained on such a biased dataset often generalize poorly when test images deviate from these dominant patterns, such as failing to recognize sea turtles without beach backgrounds. Recent research [75, 41] further shows that these biases also exist in large-scale visual datasets, including YFCC [66], CC [7], and DataComp [20], to the extent that a simple classifier can easily distinguish among datasets.

Curating datasets to mitigate such biases requires a comprehensive understanding and quantification of the distribution of visual concepts. Existing approaches typically rely on manual annotation to quantify visual concepts within large-scale datasets, but this process is costly and impractical [38, 30]. Vision-Language Models (VLMs) [39] have recently emerged as scalable tools for automatically generating descriptions of visual content. However, because these captions are expressed in natural language, they often vary in granularity and use different synonyms to describe the same content. This inconsistency makes it difficult to extract structured and meaningful visual concepts from VLM-generated captions, often requiring additional human intervention [6] or extensive prompt engineering [45], both of which present significant practical challenges.

In this paper, we introduce **ConceptScope**, a framework for analyzing visual datasets through the distribution of visual concepts, designed for image datasets with class labels (Fig. 1). ConceptScope operates in two stages: First, it constructs an interpretable concept dictionary by training Sparse Autoencoders (SAEs) [10] on image representations from vision foundation models [37]. SAEs disentangle dense representations into a sparse set of human-interpretable concepts, enabling scalable, unsupervised concept discovery without manual annotation. Second, ConceptScope characterizes the dataset by measuring each concept's semantic relevance and co-occurrence frequency with every class, and grouping them into three categories (Fig. 1 (a)): (1) *target*, essential for recognizing the class (e.g, the body or shell of a `sea turtle`); (2) *context*, frequently co-occur with the class but are not necessary for recognition (e.g., beach, underwater, or forest backgrounds); and (3) *bias*, a subset of context concepts that are class-agnostic but exhibit strong statistical correlation with specific classes (e.g., the prevalence of beach scenes for images labeled as `sea turtle`). By analyzing the distributions of these categorized concepts, ConceptScope provides class-level dataset characterizations (Fig. 1 (b)) that enable bias identification and robustness evaluation.

Through extensive experiments, we show that our method captures a broad range of visual concepts, including objects, textures, backgrounds, facial attributes, emotions, and actions. We further demonstrate that the spatial attributions derived from SAEs aligned closely with ground-truth segmentation masks. Moreover, ConceptScope reliably identifies known dataset biases (e.g., background bias in Waterbirds [59]) and uncovers previously unannotated ones (e.g., cultural biases or co-occurring objects in ImageNet [12]). Finally, we highlight its practical utility for evaluating model robustness by identifying out-of-distribution samples and quantifying the resulting degradation in model accuracy.

Our contributions are summarized as follows:

1. We introduce **ConceptScope**, a framework that systematically extracts visual concepts, categorizes them based on their relevance to target classes, and quantifies their distributions.

2. We empirically validate SAE activations as effective, interpretable concept extractors.

3. We demonstrate the practical utility of ConceptScope for detecting dataset biases and evaluating model robustness by grouping test samples into subsets having distinct concept distributions.

## 2 Related Work

**Analyze vision dataset.** A key line of research aims to comprehensively characterize visual datasets. Early works summarized datasets by analyzing local image patches [13, 57] or by selecting representative images [67]. With the advent of VLMs [39], more recent methods have leveraged VLM-generated captions to provide deeper insights. Some approaches summarize datasets by aggregating captions to highlight overall characteristics [15, 75], but they do not provide detailed, sample-level statistical

analyses of visual concepts. Other works treat captions as pseudo-labels for visual attributes, but typically rely on manually curated attribute lists [6] or extensive LLM-based prompt chaining to extract and assign attributes [32, 45]. Even after these labor-intensive steps, attributes may still be missing or inconsistently captured unless prompts are carefully crafted [8]. Our work addresses these limitations by automatically discovering semantically meaningful concepts and quantifying their sample-level distributions using SAEs.

**Bias identification via model behavior.** Several studies detect model bias by analyzing failures on test samples. These approaches segment test samples into subgroups with low performance and then assign semantic labels to the resulting clusters using cross-modal embeddings [17] (e.g., CLIP [54]) or generated captions [73, 28]. However, such methods identify dataset bias only indirectly, through downstream model behavior. If the training data is biased (e.g., birds frequently co-occurring with ocean backgrounds) but the test set lacks bias-conflicting examples (e.g., birds without ocean backgrounds), these methods cannot detect the underlying bias in the training data or the model's reliance on it. More importantly, they do not directly inspect the training data and thus cannot determine *which* training samples contain bias-inducing features or *how prevalent* those features are. Another line of work identifies biases by inspecting neuron activation maps that highlight image regions unrelated to the target objects [48, 63, 51]. However, these methods depend on manual annotation to distinguish between target-related and unrelated regions, limiting scalability. In contrast, our method operates fully automatically without requiring human intervention.

**Sparse Autoencoders (SAEs) for interpretability.** SAEs have recently emerged as a powerful tool for interpreting the internal mechanisms of deep learning models [10]. Unlike earlier approaches [52, 19, 2] that attempt to interpret individual network neurons, which often encode multiple meanings, SAEs extract latent concepts that are monosemantic and thus more interpretable [16]. Following their success in large language models [10], SAEs have been applied to vision models for interpreting classification decisions [55], explaining model adaptation [37], and analyzing diffusion models [64, 27]. Our work differs from these approaches in its primary focus: rather than interpreting model behavior, we leverage SAEs to characterize visual datasets. By disentangling visual representations from strong vision encoders such as CLIP [54] into clearly interpretable concepts, ConceptScope provides a robust foundation for understanding dataset composition and uncovering inherent biases.

## 3 ConceptScope: A Framework for Characterizing Datasets

In this section, we present **ConceptScope**, a framework for analyzing image classification datasets through the distribution of interpretable visual concepts. ConceptScope operates in two stages: (1) constructing a concept dictionary by extracting human-interpretable visual concepts by training Sparse Autoencoders (SAEs) on representation from a vision foundation model, and (2) categorizing these concepts into *target*, *context*, and *bias* types for every class, based on their semantic relevance to the class label and their frequency of occurrence in images of that class. We begin by defining the concept types and formalizing the problem setting in §3.1, then describe the construction of the concept dictionary in §3.2, and finally explain the class-wise concept categorization process in §3.3.

### 3.1 Problem formulation

Consider an image classification dataset $\mathcal{D} = \{(x_i, y_i)\}_{i=1}^N$, where each input $x_i \in \mathcal{X}$ is an image and $y_i \in \mathcal{Y} = \{1, \ldots, K\}$ is its class label among $K$ possible classes. Let $\mathcal{C} = \{c_1, c_2, \ldots, c_M\}$ denote a set of visual concepts representing recurring visual patterns such as colors, shapes, textures, backgrounds, and objects. For each class $y \in \mathcal{Y}$, we categorize concepts into three types: *target*, *context*, and *bias*.

**Target concepts.** For a class $y$, the target concepts $\mathcal{T}_y \subseteq \mathcal{C}$ contains features that are both essential and distinctive for identifying $y$ (e.g., `turtle shell` and `turtle head` for `sea turtle`). Their absence substantially hinders the correct classification of an image as belonging to the class.

**Context concepts.** For a class $y$, any non-target concept $c \in \mathcal{C} \setminus \mathcal{T}_y$ is considered a context concept. These concepts co-occur with $y$ but are not required for correct classification as long as the target concepts are present. For example, in an image of a sea turtle on a beach, the `sandy environments` and the `ocean` background are context concepts for the `sea turtle` class.

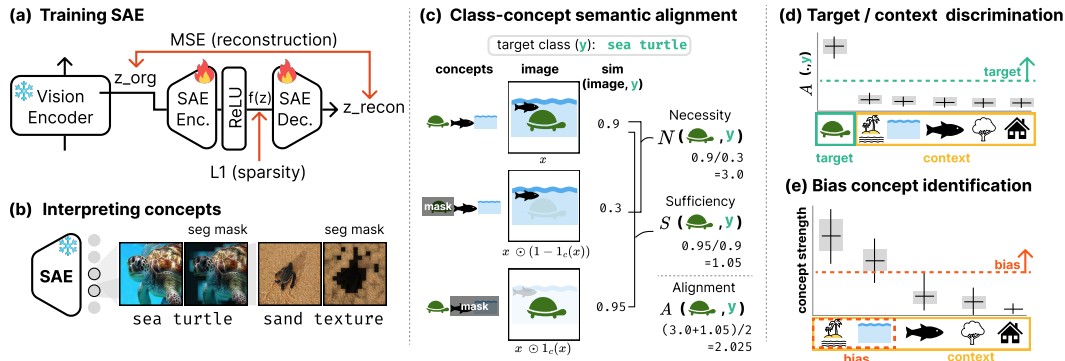

Figure 2: **ConceptScope. (a)** We train an SAE and **(b)** construct a concept dictionary to specify the semantic meaning of each latent through reference images with segmentation masks and generated textual descriptions. **(c)** We compute class-concept semantic alignment score and **(d)** categorize concepts into *target* and *context* concepts. **(e)** We further distinguish context concepts above a threshold concept strength (activation value) as *bias* concepts.

**Bias concepts.** For a class $y$, a bias concept is a context concept that disproportionately co-occurs with $y$ in the dataset. For example, if `sandy environments` appears far more frequently in images of `sea turtle` than other backgrounds such as `ocean` or `tropical scenes`, then it is considered the bias concept of that class.

Given these definitions, the goal of ConceptScope is as follows:

1. **Concept dictionary construction:** Identify a set of interpretable visual concepts $\mathcal{C}$.
2. **Concept categorization:** Categorize each concept as target, context, or bias for each class.

We describe the concept dictionary construction procedure in the next section, followed by the class-wise categorization process.

## 3.2 Constructing a concept dictionary using Sparse Autoencoders

Constructing a concept set with both broad coverage and appropriate granularity is challenging, as it requires exhaustively identifying semantically meaningful visual patterns across large datasets—a process that is labor-intensive, subjective, and prone to omission. To address this, we adopt SAEs as scalable concept extractors that automatically discover interpretable visual concepts without manual supervision. We train an SAE on representations from a strong vision encoder, such as CLIP vision transformers (ViT) [54] (Fig. 2 (a)), and interpret the learned concepts by examining their maximally activating images and identifying the shared visual patterns they capture (Fig. 2 (b)).

**Architecture of SAEs and training.** SAEs consist of a linear encoder and a decoder with a non-linear activation function [10]. Given an input image $x$, we first extract intermediate-layer token embeddings $\mathbf{z} = \{z_1, z_2, \ldots, z_l\}$ from a pretrained ViT, where $z_i \in \mathbb{R}^d$ is a patch-level token embedding, $l$ is the token length (the number of patches plus one class (CLS) token), and $d$ is the embedding dimensionality of the ViT. We train SAEs to reconstruct each of the input token embeddings $z$ through a sparse representation:

$$f(z) = \phi(W_{\text{enc}}^T z), \quad \text{SAE}(z) = W_{\text{dec}}^T f(z), \tag{1}$$

where $W_{\text{enc}} \in \mathbb{R}^{d \times d'}$ and $W_{\text{dec}} \in \mathbb{R}^{d' \times d}$ are the encoder and decoder weight matrices, respectively and $\phi$ is ReLU activation function [37]. The SAE latent dimension $d'$ is set significantly larger than $d$, using a large expansion factor (e.g, 16 or 32). We optimize the SAE by minimizing a combination of a reconstruction loss and $L_1$ sparsity penalty:

$$\mathcal{L} = ||z - \text{SAE}(z)||_2^2 + \lambda ||z||_1, \tag{2}$$

where the $\lambda$ controls the degree of sparsity in the latent representation (Fig. 2 (a)). We utilize the pretrained CLIP-ViT-L model with a patch size of 14 and train the SAE using ImageNet-1K [12]. Further implementation and training details are provided in the Appendix §B.1.

**Quantifying concept activations.** After training, the SAE provides fine-grained quantification of concept activations at both patch and image levels. Each row of the decoder weight matrix $W_{\text{dec}}$

corresponds to a linear representation of a learned concept, while the encoder output $f(z)$ represents the activation strength of each concept for a patch embedding $z$. These patch-level activations can be visualized as coarse segmentation masks, localizing concepts within images (§4.2). For image-level analysis, we aggregate patch-level activations by averaging the encoder outputs across all patch tokens: i.e., $f(\mathbf{z}) = \frac{1}{l} \sum_{z_i \in \mathbf{z}} f(z_i); f(\mathbf{z}) \in \mathbb{R}^{d'}$. The scalar value $f(\mathbf{z})_c$ then represents the overall activation strength of concept $c$ for the image.

**Interpreting learned concepts.** To interpret the semantics of each latent, we collect the top-activating images per concept and extract their corresponding segmentation masks, following the procedure introduced in Lim et al. [37]. We then use multimodal large language models (GPT-4o) to generate short natural language descriptions from these visual inputs (Fig. 2 (b)). More examples are illustrated in Fig. 6. It is worth noting that this annotation step is fully automated, except for light prompt tuning, and intended purely for human interpretability. The generated descriptions are not used in the concept categorization procedure described in §3.3. In Appendix B.2, we provide further details, including prompts, implementation specifics, and an evaluation of annotation quality.

### 3.3 Categorizing concepts as target, context, and bias

After building the concept dictionary, we categorize concepts for each class. Given a labeled dataset of (image, class) pairs, we first compute a *class–concept alignment score*, which measures how representative a concept is of a class and how essential it is for recognizing that class (Fig. 2 (c)). Based on this score, we separate concepts into target and context categories (Fig. 2 (d)). We then further divide the context concepts into bias and non-bias categories using *concept strength*, which measures how frequently and confidently a concept appears within a class (Fig. 2 (e)).

**Separating target and context concepts via alignment scores.** To distinguish **target** from **context** concepts for a class $y$, we define a *class–concept alignment score*. Let $\mathcal{C} = \{c_1, \ldots, c_M\}$ be the set of learned concepts. For each concept $c$, we compare the model's prediction confidence under three conditions: (1) using all concepts $\mathcal{C}$, (2) excluding $c$ ($\mathcal{C} \setminus \{c\}$), and (3) using only $c$ ($\{c\}$). We obtain the set $\mathcal{C} \setminus \{c\}$ by masking out the spatial attribution[2] of $c$ from the image $x$ ($x \odot (1 - m_c(x))$), and $\{c\}$ by keeping only concept region ($x \odot m_c(x)$).

We formalize this comparison with two metrics: *necessity* $N(c, y)$ and *sufficiency* $S(c, y)$. Necessity measures the drop in prediction confidence when the $c$ is removed (i.e., case (1) vs. (2)), while sufficiency measures the retained confidence when only $c$ remains (i.e., case (1) vs. (3)) (Fig. 2 (c)):

$$\text{N}(c, y) = \frac{1}{|X_y|} \sum_{x \in X_y} \frac{P(y \mid x)}{P(y \mid x \odot (1 - m_c(x)))}, \quad \text{S}(c, y) = \frac{1}{|X_y|} \sum_{x \in X_y} \frac{P(y \mid x \odot m_c(x))}{P(y \mid x)}, \quad (3)$$

where $X_y$ is the set of images belonging to class $y$, $P(y \mid x)$ is the model's confidence for class $y$ given $x$, computed as the cosine similarity between the CLIP text embedding of $y$ and the image embedding of $x$, $m_c(x)$ is the binary mask for concept $c$, and $\odot$ denotes element-wise multiplication.

We combine the necessity and sufficiency metrics into a single alignment score $A(c, y) = \frac{\text{N}(c,y)+\text{S}(c,y)}{2}$. A concept $c$ is labeled as **target** if $A(c, y) \geq \mu_y^{\text{align}} + \alpha \times \sigma_y^{\text{align}}$, and as **context** otherwise (Fig. 2 (d)). Here, $\mu_y^{\text{align}}$ and $\sigma_y^{\text{align}}$ are the mean and standard deviation of alignment scores for class $y$, and $\alpha$ is a threshold factor. In Appendix B.3, we provide further details, empirical validation, and illustrative examples (Fig. 7).

**Distinguishing bias and context concepts via concept strength.** We measure concept prevalence using the SAE encoder output $f(\mathbf{z})$ (§3.2), which is non-negative by design. For each concept $c$, we define its *concept strength* for class $y$ as the average activation across all samples $\mathbf{z} \in Z_y$ belonging to class $y$: i.e., $\tilde{f}_{c,y} = \text{avg}_{\mathbf{z} \in Z_y}(f(\mathbf{z})_c)$. A high concept strength indicates that the $c$ is frequently and confidently associated with the class $y$. After excluding the target concepts $\mathcal{T}_y$, we compute the mean $\mu_{c',y}^{c;s} = \text{avg}_{c'}(\tilde{f}_{c',y})$ and standard deviation $\sigma_{c',y}^{c;s} = \text{std}_{c'}(\tilde{f}_{c',y})$ of concept strengths for all remaining context concepts $c' \in \mathcal{C} \setminus \mathcal{T}_y$. We classify $c$ as a **bias** concept if its strength exceeds one standard deviation above the mean ($\tilde{f}_{c,y} \geq \mu_{c',y}^{c;s} + \sigma_{c',y}^{c;s}$); otherwise we classify it as a non-bias context concept (Fig. 2 (e)).

---

[2]We validate the accuracy of these spatial attributions in §4.2.

Table 1: **Performance comparison for concept prediction**. We compare our SAE-based method in ConceptScope with generated caption-based (BLIP-2, LLaVA-NeXT) methods and show that our proposed approach outperforms the baselines. We use six labeled datasets and report class-wise binary classification accuracy in $F_1$ and AUPRC, averaged across classes. Values represent the mean and standard deviation across classes.

| Method | Metric | Caltech101 [18] (Objects) | DTD [9] (Textures) | Waterbird [59] (Backgrounds) | CelebA [43] (Facial Attr.) | RAF-DB [36] (Emotions) | Stanford40 [72] (Actions) | Average |
|---|---|---|---|---|---|---|---|---|
| BLIP-2 [35] | $F_1$ | 0.64±0.35 | 0.38±0.25 | 0.37±0.10 | 0.27±0.24 | 0.24±0.17 | 0.66±0.18 | 0.43 |
| LLaVA-NeXT [34] | $F_1$ | 0.61±0.35 | 0.40±0.21 | 0.57±0.12 | 0.62±0.24 | 0.45±0.18 | 0.80±0.16 | 0.58 |
| **ConceptScope** | $F_1$ | 0.83±0.21 | 0.57±0.20 | 0.78±0.07 | 0.81±0.11 | 0.55±0.18 | 0.78±0.13 | 0.72 |
| **(SAE)** | AUPRC | 0.89±0.19 | 0.57±0.23 | 0.83±0.09 | 0.85±0.13 | 0.59±0.21 | 0.82±0.15 | 0.76 |

**Analyzing class-wise concept distributions.** By analyzing the distribution of target, context, and bias concepts, ConceptScope allows practitioners to evaluate the diversity and dominance of visual concepts for each class. Specifically, the distribution of target concepts reflects the range of visual appearances and prototypical features that define the class, while variation among context concepts captures the diversity in backgrounds and co-occurring objects (Fig. 8, Fig. 9). The concept strength of bias concepts highlights the dominance of spurious patterns, revealing potential shortcuts a model may exploit during training (§5).

## 4 Sparse Autoencoders are Reliable Concept Extractors

In this section, we verify that SAEs capture a diverse set of visual concepts and that their activation values are reliable indicators of concept presence. We first evaluate the predictive capability of SAE latent activations through binary attribute classification in §4.1. Subsequently, we assess the accuracy of segmentation masks derived from SAE activations in §4.2.

### 4.1 Evaluating concept prediction performance

**Tasks.** We perform binary attribute classification to validate that SAE activations accurately reflect the presence of visual concepts. Evaluation is conducted across six annotated datasets covering diverse visual attributes: Caltech101 [18] (objects), DTD [9] (textures), Waterbird [59] (backgrounds), CelebA [43] (facial attributes), RAF-DB [36] (emotions), and Stanford40 Actions [72] (actions). Additional dataset details are provided in Appendix §C.1.

**Methods.** For each test sample, we obtain the image-level activation values $f(\mathbf{z}) \in \mathbb{R}^{d'}$ and treat them as unbounded confidence scores for concept presence. For each target attribute, we identify the most relevant latent dimension from the training split that maximizes the Area Under the Precision–Recall Curve (AUPRC), and determine the prediction threshold that maximizes the $F_1$ score. We reported AUPRC, the $F_1$ score, precision, and recall as final metrics.

**Baselines.** Recent approaches [45, 32] use VLMs to generate captions and apply extensive prompt chaining with LLMs to assign attribute labels. To minimize dependence on LLMs, we implement a simplified caption-based baseline: for each generated caption, we query an LLM to decide whether it implies the presence of the target attribute. We evaluate two state-of-the-art open-source VLMs, BLIP-2 [35] and LLaVA-NeXT [34], and report their $F_1$ scores, precisions, and recalls for direct comparison with our SAE-based results. The details of baselines are provided in Appendix §C.2.

**SAE captures diverse visual concepts.** As shown in Table 1, SAE outperforms baselines, achieving an $F_1$ of 0.72 and AUPRC of 0.76 with low variation across four random seeds (std. 0.01). Caption-based methods (BLIP-2, LLaVA-NeXT) perform worse due to linguistic variability, often using generic terms (e.g., simplifying "dotted pattern" with "pattern"). In contrast, SAE directly encodes visual concepts through latent features, consistently outperforming baseline methods across diverse visual attributes. Additional metrics are reported in Table 6, precision–recall curves are shown in Fig.10, and representative failure cases of caption-based methods are illustrated in Fig.11.

**SAE activation is highly correlated with concept presence.** We further evaluate whether SAE activation strength reflects the degree of the presence of visual concepts. To this end, we treat the similarity between CLIP image embeddings and textual embeddings of class labels as pseudo-ground truth and assess their correlation with the image-level SAE activation values. Table 7 summarizes

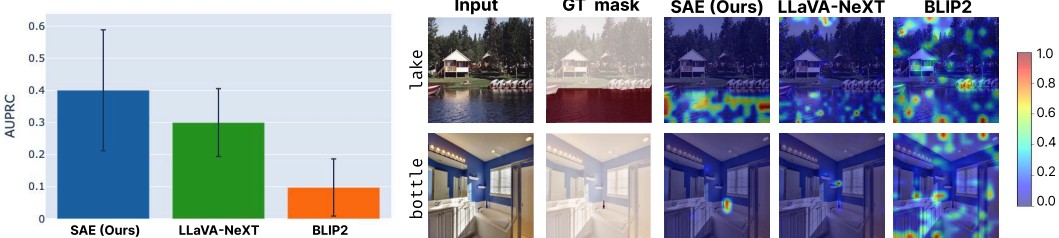

Figure 3: **SAE provides reliable segmentation masks.** Quantitative and qualitative comparisons of SAE spatial attribution (ours) with attention map-based approaches from BLIP-2 [35] and LLaVA-NeXT [34] on ADE20K [77]. We report the Area Under Precision-Recall Curve (AUPRC) of segmentation masks. SAE results are averaged over four models trained with different random seeds, yielding a standard deviation of 0.002. Error bars indicate the standard deviation across 150 classes.

the Pearson ($r$) and Spearman ($\rho$) correlation coefficients across the six attribute-labeled datasets. We observe consistently strong correlations ($r = 0.71$, $\rho = 0.65$), confirming that SAE activations reliably reflect visual concept presence. Further details are provided in Appendix §C.4.

## 4.2 Evaluating concept localization

**Tasks.** Since we utilize the spatial attribution of SAE activations to distinguish between target and context concepts, we evaluate the localization accuracy of these activations. Specifically, we conduct binary segmentation experiments on the ADE20K [77] dataset, a widely-used benchmark with pixel-level annotations across 150 semantic categories (e.g., people, animals, sky, and buildings). Evaluation is conducted on 2,000 images from the ADE20K validation set.

**Methods.** We use patch-level SAE activation for a concept $c$, $f(z)_c$ (§3.2) as coarse segmentation masks, and normalize the value into a range of $[0, 1]$. We discretize these continuous-valued SAE activations at varying thresholds and measure AUPRC. Following the approach in §4.1, we assign each latent to the class for which it achieves the highest AUPRC on the training split.

**Baselines.** We compare against attention maps derived from VLMs, as they provide spatial localization with respect to textual input. Specifically, we use BLIP-2 [35] and LLaVA-NeXT [34], state-of-the-art VLMs evaluated in §4.1. We obtain the attention maps by taking the max value across heads and averaging across layers of self-attention weights of image tokens.

**SAE generates reliable segmentation masks.** Fig. 3 compares the segmentation performance of SAE with attention-based masks from LLaVA-NeXT and BLIP-2. SAE achieves the highest AUPRC (BLIP-2: 0.098, LLaVA-NeXT: 0.302, Ours: 0.399), performing particularly well on large regions (e.g., "lake"), where activations closely match ground-truth masks. In contrast, LLaVA-NeXT maps contain noise, and BLIP-2 maps include irrelevant activations. Moreover, SAE efficiently produces segmentation masks for all concepts simultaneously in a single forward pass, whereas the VLM-based methods require specifying each target class and multiple inference passes per image. Although performance drops for small objects due to coarse $16 \times 16$ resolution, SAE still localizes them reliably (e.g., the "bottle" example in Fig. 3), supporting its use for computing alignment scores in §3.3.

## 5 ConceptScope for Dataset and Model Bias Analysis

In this section, we demonstrate the capability of ConceptScope to discover dataset biases. We first perform quantitative evaluation on datasets with annotated bias attributes and show that our approach accurately detects known biases (e.g., background bias from Waterbirds [59]) in §5.1. We then extend our analysis to real-world datasets such as ImageNet-1K [12], Food101 [4], and SUN397 [69], where bias attributes are not explicitly annotated, revealing previously unrecognized biases in §5.2. Lastly, we show an application of ConceptScope for diagnosing model robustness under concept distribution shifts in §5.3

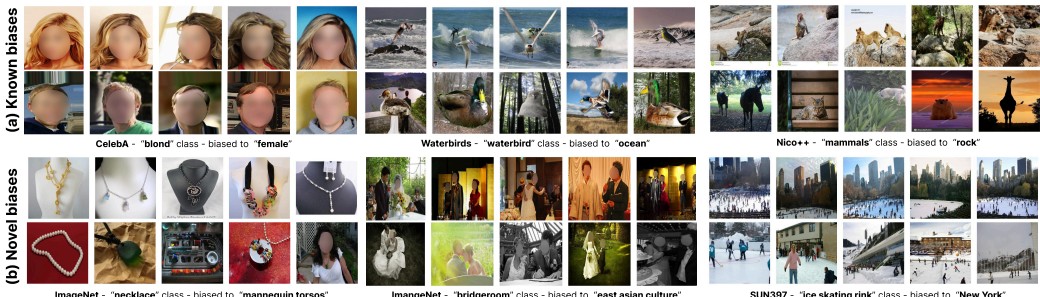

Figure 4: **ConceptScope discovers dataset biases.** We discover **(a)** known and **(b)** novel biases from bias attribute annotated and unannotated datasets, respectively. The top row of each panel shows bias-*aligned* examples, and the bottom row shows examples *without* bias attributes.

Table 2: **ConceptScope outperforms training-based baselines in bias discovery** (Precision@10). For NICO++, values in parentheses represent bias severity levels, with higher values indicating stronger correlations between classes and spurious attributes in the training dataset.

| Method | Waterbirds [59] | CelebA [43] | Nico++(75) [76] | Nico++ (90) [76] | Nico++ (95) [76] |
|---|---|---|---|---|---|
| DOMINO [17] | 90.0% | 87.0% | 24.0% | 24.0% | 24.0% |
| FACTS [73] | 100.0 % | 100.0% | 55.0% | 60.8% | 61.0% |
| ViG-Bias [46] | 100.0% | 100.0% | 60.0% | 66.7% | 65.0% |
| **ConceptScope (Ours)** | **100.0%** | **100.0%** | **72.9%** | **73.1%** | **74.0%** |

## 5.1 Detecting known biases

**Tasks.** We evaluate our framework on the bias discovery task introduced by Yenamandra et al. [73]. In this setup, each class in the training set is associated with a distinct spurious attribute, such as a background scene. At test time, images of a target class $y$ are divided into subsets, each characterized by a visual attribute spuriously correlated with a different class $y'$. The goal is to correctly identify the spurious attribute associated with each subset. Performance is measured using Precision@10 [17], where we retrieve the top 10 most strongly associated with the candidate attribute and compute the proportion that indeed contain the ground-truth spurious attribute. We conduct experiments on three widely used benchmarks: CelebA [43], Waterbirds [59], and NICO++ [76]. Example images are shown in Fig. 4 (a). Details regarding datasets and evaluation protocols are provided in Appendix §D.1.

**Methods and baselines.** Our approach first identifies bias concepts within each training class. For each class, we extract SAE latent activations corresponding to the bias concepts from other classes and retrieve the top 10 images with the highest activations from the test set. We compare ours against three baselines: DOMINO [17], FACTS [73], and ViG-Bias [46]. Further details regarding baselines are provided in the Appendix §D.3.

**ConceptScope accurately detects known dataset biases.** Table 2 shows that our proposed framework achieves state-of-the-art performance on the bias discovery task, substantially outperforming existing baselines. Reported results of ConcepScope are averaged over four independently trained SAEs, with a standard deviation below 0.02, confirming the robustness of our approach. Unlike baseline methods that rely on classifier training and failure-mode analysis, ConceptScope directly uncovers bias concepts from the structure of the training dataset. Moreover, once the SAE is trained, it can be applied to analyze biases in other datasets without retraining, leveraging the strong transferability of SAEs [37]. Finally, our method provides localization of spurious attributes through segmentation masks, clearly highlighting the input regions responsible for the bias activation (Fig. 12).

## 5.2 Discovering novel biases in the wild

**Biases in real-world datasets.** Using our framework, we discover previously unrecognized biases in real-world datasets. As shown in Fig. 4 (b), we identify various types of biases across ImageNet-1K and SUN397. We detect unexpected object associations (e.g., "necklaces" frequently appearing with "mannequins"), cultural biases (e.g., the class "bridegroom" correlated with "East Asian cultural" con-

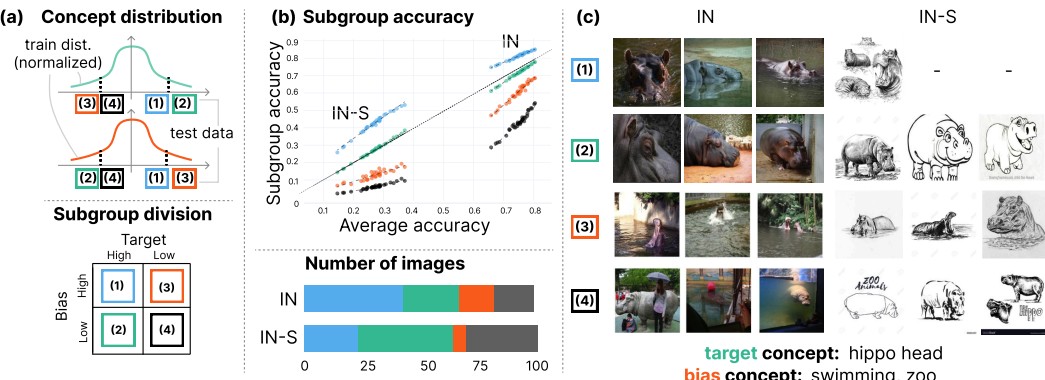

Figure 5: **Diagnose model robustness. (a)** We divide test data into four subgroups (1-4) by comparing its concept strength (high or low) to the training distribution: (h, h), (h, l), (l, h), and (l, l) for target and bias concepts. **(b)** We use ImageNet-1K (IN) and ImageNet-Sketch (IN-S) for the analysis and observe groups (1) & (2) are the majority in both datasets, and their group-wise accuracy of 34 pretrained vision models (e.g., ResNet, ViTs) shows a consistent tendency: from group (1) to (4) in descending order. **(c)** Example images categorized in each subgroup for class "hippo".

texts), and location-specific biases ("ice skating rink" correlated with "New York City"). Additional examples, including those from the Food101, are shown in Fig. 13, with further details provided in Appendix §D.5. Across these datasets, the average number of bias concepts per class is 2.45, indicating that ConceptScope effectively captures multiple biases within a single class (Fig. 7).

## 5.3   Diagnosing Model Robustness

In this section, we demonstrate the application of our proposed framework, ConceptScope, as a diagnostic tool for model robustness against concept distribution shifts. Model robustness is typically evaluated by measuring the performance drop on distribution-shifted test sets, such as from ImageNet [12] to ImageNet-V2 [56] or ImageNet-Sketch [68]. However, in many domains (e.g., food or scene classification), such curated out-of-distribution (OOD) benchmarks are unavailable, and constructing them manually can be both costly and time-consuming. ConceptScope provides a complementary approach that enables robustness assessment directly within the original test set, without requiring additional OOD datasets.

**Approach.** Using target and bias concepts identified by ConceptScope, we partition the test set of each class into four subgroups: (1) high target-high bias, (2) high target-low bias, (3) low target-high bias, and (4) low target-low bias (Fig. 5 (a)). A test sample is classified as having 'high' concept strength if its activation exceeds the average activation of that concept across all training images in the corresponding class. These groups can be interpreted as forming a spectrum of distribution shift. Group 1 represents the most typical setting seen during training, where both the target and bias concepts co-occur. In contrast, Group 4 represents a more unusual or outlier scenario, where neither the target nor the bias concepts are present, posing a stronger generalization challenge for the model.

**Experimental setup.** We compute concept distributions from the training split of ImageNet-1K, then partition the ImageNet validation set and the ImageNet-Sketch test set into distinct groups accordingly. We subsequently evaluate and compare model accuracy across these groups across 34 model weights trained on ImageNet-1K, including diverse architectures such as VGG [62], ResNet [23], and Transformer-based models [14]. The complete list of models is provided in Appendix §E.

**Key findings.** In Fig. 5 (b), the y-axis represents subgroup accuracy (with the four groups shown in different colors), and the x-axis indicates the average accuracy across all four subgroups. Each vertical line corresponds to a single model, with four colored dots representing that model's accuracy on each subgroup. Among the 34 models evaluated, ConvNeXt-Large [42] achieves the highest average accuracy and is shown on the rightmost line (in both ImageNet (IN) and ImageNet-Sketch (IN-S)). Notably, it also performs best on Group 4, the most challenging subgroup. We observe that models with higher accuracy on Group 1 (the most frequent training setting) also tend to perform better on Group 4 (the rare or outlier setting). This trend is consistent with prior findings [47], which

show that out-of-distribution (OOD) performance often correlates with in-distribution performance. Crucially, our approach enables such robustness analysis without requiring an external OOD test set, using only concept-aware subgroups derived from the original data. Interestingly, we observe the same pattern on the ImageNet-Sketch, despite its large domain shift and generally lower performance. Representative examples from each group are visualized in Fig. 5 (c).

# 6 Discussion

**Concepts beyond ImageNet.** Although the SAE is trained on ImageNet, our SAE-derived concepts generalize to other datasets used in CLIP zero-shot evaluation, such as Food101 [4] and SUN397 [69]. This generalization is expected, as the SAE is not trained using ImageNet class labels. Instead, it learns to reconstruct image representations from a general-purpose foundation model, which is assumed to have already captured diverse visual concepts, as evidenced by its strong zero-shot performance. For datasets that differ substantially from ImageNet, such as medical images, the SAE should be trained on the target domain and/or paired with a more suitable backbone model. Prior work, including CytoSAE [11] and Mammo-SAE [50], has shown that SAE can be successfully applied to medical images, and a similar approach can be used for domain-specific analysis.

**Extension to multi-label settings.** To examine its applicability beyond single-label classification, we apply ConceptScope to a multi-label setting using the MSCOCO 2017 test set [38]. We extract object mentions from ground-truth captions and treat them as multi-label targets and conduct concept categorization as in Section 3.2. The analysis reveals distinct bias patterns: *cat* is often associated with indoor environments, *dog* with couches and living rooms, and *bird* with bird feeders and trees. These findings indicate that ConceptScope can detect meaningful bias concepts even in complex, real-world multi-label datasets.

**Comparison with existing approach for bias analysis in vision datasets.** To further contextualize our framework, we additionally review prior work on bias analysis in vision datasets. LaViSE [71] interprets the semantic meaning of individual neurons and uses this to examine gender bias in the MS-COCO dataset [38]. Similarly, SpLiCE [3] decomposes CLIP representations into sparse, human-interpretable concepts based on a predefined concept set and analyzes bias between the `Woman` and `Man` classes of CIFAR-100 [31]. While these approaches provide valuable insights into dataset bias, they are limited to predefined gender categories, implicitly treating any unrelated concept as bias. For instance, such methods require manual specification of what constitutes bias (e.g., considering `sand environment` as a bias for `sea turtle`). In contrast, ConceptScope automatically determines whether a concept is bias or not based on the semantics relationship to the class label, removing the need for manually defined bias categories. Further analyses, including detailed comparisons with existing approaches to segmentation-based concept discovery, are provided in the Appendix A.

# 7 Conclusion

**Limitations.** Our SAE-discovered concepts are constrained by the knowledge encoded in CLIP's representations. For example, concept prediction scores (Table 1) on domain-specific datasets such as RAF-DB [36] (emotion) or DTD [9] (texture) are lower compared to more general datasets. Leveraging vision foundation models fine-tuned on these domain-specific datasets could yield more concrete and task-relevant latent concepts. Additionally, our segmentation masks are patch-level and coarse, leaving room for improvement in localization accuracy. Despite these limitations, our concept-level analysis still uncovers previously unrecognized, non-trivial biases.

**Conclusion.** We introduced ConceptScope, a framework that uses SAE to detect and characterize biases in image classification datasets. Our pipeline automatically builds a concept dictionary, categorizes concepts into target, context, and bias, and enables detailed dataset analysis. Experiments show that SAEs are reliable concept extractors, surpassing prior methods and achieving state-of-the-art results in bias discovery benchmarks. ConceptScope also uncovers new biases in complex datasets such as ImageNet and can diagnose model robustness without external out-of-distribution datasets.

## Acknowledgements

This work was supported by Institute for Information & communications Technology Planning & Evaluation (IITP) grant funded by the Korea government (MSIT) (RS-2019-II190075, Artificial Intelligence Graduate School Program(KAIST)), the National Research Foundation of Korea (NRF) grant funded by the Korea government(MSIT) (No. RS-2025-00555621), and Institute of Information & communications Technology Planning & Evaluation (IITP) grant funded by the Korea government (MSIT) (No.RS-2021-II212068, Artificial Intelligence Innovation Hub).

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

# Appendix

# A  Additional Discussion

**Concept discovery based on segmentation.** Recent advances in segmentation models, such as SAM [29], have achieved impressive generalization performance, motivating several studies to use segmentation for concept discovery. HU-MCD [21], for example, combines SAM-generated region masks with feature clustering to extract object- or background-level concepts. Although HU-MCD demonstrates potential in identifying high-level visual entities (e.g., object parts, background elements), it has notable limitations. It struggles to capture fine-grained attributes such as textures, colors, or object subtypes, and lacks a formal mechanism to quantify how each discovered concept relates to class identity or dataset bias. HU-MCD typically identifies coarse-level object or background concepts (e.g., "airplane", "airplane tail", "sky"), whereas the Sparse Autoencoders used in ConceptScope uncover more granular and semantically diverse concepts such as "commercial airplanes", "military aircraft", and "jets", as well as state-dependent variations like "landing" and "flying".

**On the Role of Segmentation in ConceptScope.** Unlike traditional segmentation tasks, producing high-quality masks is not the primary objective of our framework. The segmentation derived from SAE spatial attributions is used to compute the alignment score, enabling ablation or isolation of each concept within an image. Our objective is to analyze concept–class relationships and categorize concepts into target, context, and bias types, rather than optimizing segmentation quality. Despite imperfect segmentation performance, ConceptScope effectively reveals meaningful real-world biases, and its interpretability can be further enhanced through integration with external segmentation models.

# B  Method Details: ConceptScope

In this section, we provide additional technical details and illustrative examples for our proposed framework ConceptScope, supplementing §3 of the main paper.

## B.1  SAE training

We include details regarding hyperparameters and computational resources for training the SAE, supplementing the explanations provided in §3.2 of the main paper.

**Hyperparameter settings.** We train the SAE using OpenAI CLIP ViT-L/14[3] as the vision encoder and extract embeddings from its penultimate (23rd) layer with embedding dimension $d = 1024$. The vision encoder resizes the input images to $224 \times 224$, producing 257 tokens consisting of one class token ([CLS]) and $16 \times 16$ spatial tokens. We apply an expansion factor of 32, resulting in a SAE latent dimensionality $d' = 32,768$. We set the $L_1$ sparsity loss weight $\lambda = 8 \times 10^{-5}$, and the learning rate to $4 \times 10^{-4}$ with 500 warmup steps using constant scheduling. We initialized decoder bias with the geometric median [53] of the first training batch and applied ghost gradients [5] to resample dead neurons caused by sparsity regularization. We use a batch size of 64 and train on ImageNet-1K [4] training split, comprising approximately 1.28 million images, for 5 epochs.

**Computation resources.** We ran all experiments on a single NVIDIA RTX A6000 GPU. With precomputed CLIP image embeddings, each iteration (batch size 64) took about 0.79 seconds, used 24 GB of GPU memory, and required 2,208 GFLOPs.

## B.2  Concept dictionary

We provide implementation details for constructing the concept dictionary introduced in §3.2 of the main paper, including the filtering of meaningful latent dimensions and the use of multimodal Large Language Models (LLMs) to assign concept names.

**Latent filtering.** As noted by Lim et al. [37], not all SAE latent dimensions are interpretable or monosemantic. In some cases, we observe that multiple unrelated concepts activate the same latent. To improve interpretability, we filter out uninformative or entangled latents and retain only interpretable ones for analysis. We first compute image-level activation values $f(\mathbf{z})_c$ for all latent

---

[3]https://huggingface.co/openai/clip-vit-large-patch14
[4]https://huggingface.co/datasets/evanarlian/imagenet_1k_resized_256

dimensions across the ImageNet training set and apply two filtering criteria. First, we discard latents whose maximum activation never exceeds 0.5, since we empirically observed that activations above 0.5 consistently represent clear concepts, whereas lower values fail to do so. Second, we discard latents whose concept strength exceeds 0.1, because they activate too frequently across many images and therefore lack interpretability. After filtering, we retain 3,103 latent dimensions, which is about 10% of the original set.

**Latent interpretation.** To assign semantic labels to latent dimensions, we collect the top-activating image patches for each concept and extract their corresponding segmentation masks, following the standard procedure introduced in Lim et al. [37]. Using the top five images and their masks, we query the GPT-4o API[5] alongside the following prompt:

---

Prompt

```
Identify the shared concept among images, focusing solely on non-blocked
areas and excluding dark silhouette areas, using 2-3 clear and specific words.
Answer with concepts only.
```

---

Annotating all 3,103 latent dimensions costs approximately $7 USD in total.

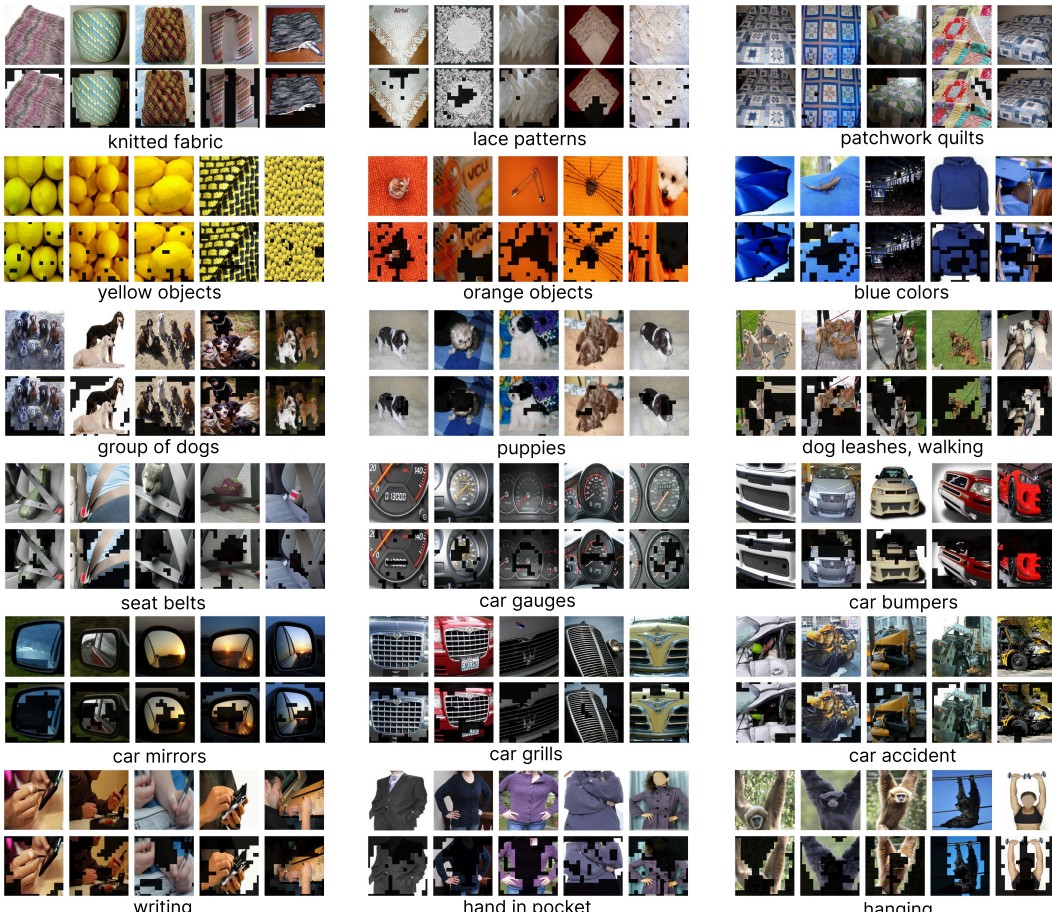

Figure 6: **Examples of discovered concepts.** The top row shows the five images from ImageNet with the highest activations for each latent dimension, while the bottom row shows the corresponding segmentation mask-applied images. The concepts identified include various colors, textures, objects, and actions. The assigned semantic labels are generated by GPT-4o.

---

[5]https://platform.openai.com/docs/models/gpt-4o

Table 3: Examples of concept descriptions generated by LLaVA-NeXT and GPT-4o, along with their similarity scores computed in CLIP and OpenAI text embedding spaces.

| LLaVA-NeXT | GPT-4o | CLIP Similarity | OpenAI Similarity |
|---|---|---|---|
| curtains | curtains | 1.00 | 1.00 |
| rooster | red rooster | 0.81 | 0.93 |
| vehicles | convertible cars | 0.76 | 0.88 |
| textured | knitted fringe | 0.53 | 0.81 |

**Examples of discovered concepts.** In Fig. 6, we visualize examples of the discovered concepts along with segmentation masks for the five most highly activated images from ImageNet for each concept. These latents captures a diverse range of visual concepts, encompassing colors, textures, objects, and actions. Additionally, the discovered concepts capture fine-grained distinctions within broader categories. For instance, car-related concepts include detailed components such as seat belts, gauges, bumpers, rear-view mirrors, grills, and even scenes depicting car accidents. Similarly, dog-related concepts captures specific scenarios, including groups of dogs, puppies, and dog leashes. Human action concepts include writing, placing hands in pockets, and hanging objects.

**Generated annotation quality.** Qualitative inspection shows that the generated annotations are meaningful across similar visual patterns, producing concise and interpretable phrases such as "knitted fabric" or "yellow objects" (Fig. 6). To verify that the top five activating images for each concept depict coherent patterns and that the resulting descriptions are stable across multimodal LLMs, we compared captions generated by GPT-4o and LLaVA-NeXT[6] using the same prompt. The average cosine similarity between the two sets of captions was 0.754 ($\pm$0.119) in CLIP's text embedding space[7] and 0.862 ($\pm$0.050) in OpenAI's text-embedding-ada-002 space[8], indicating strong agreement and convergence toward simple, intuitive descriptions. Examples of caption pairs are provided in Table 3.

## B.3 Concept categorization: target, bias, and context

In this section, we provide implementation details and empirical validation of the concept categorization methods, complementing the description in §3.3 of the main paper.

**Implementation details for computing alignment scores.** As explained in §3.3, we categorize each class's concepts into *target* and *context* concepts by measuring the differences in cosine similarity between class labels and concept-ablated images. To compute text embeddings, we use only the class name without any additional textual prompts. For each class, we randomly sample 128 training images and calculate sufficiency and necessity scores (Eq. 3 in the main paper). Empirically, we observe that the top 20 concepts per class exhibit significantly higher concept strength, while the remaining concepts contribute negligibly. We therefore compute alignment scores only for these top 20 latents.

### B.3.1 Robustness of alignment score to potential bias in CLIP

CLIP similarity scores are widely used in practice, including for data filtering [60] and image caption quality assessment [24]. Although CLIP may carry biases from its pretraining data, our method is robust to such potential biases by designing the alignment score to rely on the relative change in similarity when specific concepts are ablated or isolated, rather than on absolute similarity values.

**Toy experiment - setup.** To demonstrate this robustness, we designed a simple toy experiment. The concept *ocean* is a potential bias for the class *albatross* (a seabird), as they often co-occur. Using the Waterbirds dataset [59], we sampled 100 *albatross* images and applied segmentation masks to isolate the bird from its background. We then composited the segmented birds onto either ocean or forest backgrounds, producing two groups of 100 images each.

**Toy experiment - results.** The CLIP similarity scores between these two groups and the text prompt *albatross* were 0.281 and 0.252, respectively. As expected, the absolute scores suggest that the

---

[6]https://huggingface.co/llava-hf/llama3-llava-next-8b-hf
[7]https://huggingface.co/openai/clip-vit-large-patch14
[8]https://platform.openai.com/docs/models/text-embedding-ada-002

CLIP slightly favors *ocean* background than the *forest* background for the *albatross* class. If our alignment scores were sensitive to this bias, it might incorrectly classify *ocean* as a target rather than a context concept. However, as shown in the table below, the alignment scores for *ocean* and *forest* are consistently lower than those of the *seabird* concept in both background settings, which correctly categorizes them as context concepts. This confirms that our approach is robust to potential CLIP bias.

|  | seabird | ocean | forest |
|---|---|---|---|
| **ocean background** | **1.05** | 0.83 | 0.85 |
| **forest background** | **1.15** | 0.88 | 0.84 |

For reference, the average alignment score across datasets is $1.01 \pm 0.05$ for target concepts and $0.87 \pm 0.04$ for context concepts, supporting the correct classification in this example.

**Consistency across CLIP variants.** To further assess the robustness of our method to CLIP's internal biases, we recompute alignment scores using three different CLIP models trained on diverse datasets: (1) OpenAI CLIP[9] (used in our main experiments) (2) CLIP trained on LAION-2B[10] (3) CLIP trained on DataComp[11]. We evaluate the consistency of concept categorization across these models on Food101 [4], SUN397 [69], and ImageNet [12] datasets. Using Cohen's kappa [33] to measure agreement, we obtain an average pairwise score of 0.875, indicating almost perfect agreement. The standard deviation of alignment scores across models is only 0.110, suggesting that our categorization remains stable despite potential differences in model bias.

### B.3.2 Empirical validation of thresholds

**Empirical validation of alignment score threshold.** We observe that per-class alignment score distributions typically form two clusters: high and low, corresponding to target and context concepts. To separate them, we apply a threshold defined as the mean plus $\alpha \times$ standard deviation. We select $\alpha$ as the value that maximizes the silhouette score [58], a standard clustering metric that quantifies how well each point fits within its assigned cluster compared to other clusters. In Table 4, we report the silhouette score [58] for six datasets: Waterbirds [59], CelebA [43], Nico++ [76], ImageNet [12], Food101 [4], and SUN397 [69]. In most cases, the silhouette score peaks near $\alpha = 0$, while some datasets, such as ImageNet, achieve slightly higher scores at $\alpha = 1$.

Table 4: Average silhouette scores ($\pm$ standard deviation) for different $\alpha$ values across datasets. Each cluster is formed by grouping target and context concepts.

| Dataset | $\alpha = -3$ | $\alpha = -2$ | $\alpha = -1$ | $\alpha = 0$ | $\alpha = 1$ | $\alpha = 2$ | $\alpha = 3$ |
|---|---|---|---|---|---|---|---|
| Waterbirds | 0.00 | 0.00 | 0.00 | 0.74 | **0.79** | 0.56 | 0.00 |
| CelebA | 0.00 | 0.26 | 0.42 | **0.53** | 0.43 | 0.23 | 0.00 |
| Nico++ | 0.00 | 0.00 | 0.36 | **0.50** | 0.28 | 0.00 | 0.00 |
| ImageNet | 0.00 | 0.01 | 0.21 | 0.59 | **0.61** | 0.48 | 0.00 |
| Food101 | 0.00 | 0.00 | 0.29 | **0.60** | 0.55 | 0.30 | 0.05 |
| SUN397 | 0.00 | 0.05 | 0.34 | **0.56** | 0.55 | 0.37 | 0.06 |
| **AVG** | 0.00 ($\pm$0.00) | 0.05 ($\pm$0.10) | 0.27 ($\pm$0.15) | **0.59 ($\pm$0.08)** | 0.54 ($\pm$0.10) | 0.32 ($\pm$0.20) | 0.02 ($\pm$0.03) |

**Empirical validation of concept strength threshold.**

Concept strength distributions typically include a small number of outliers representing concepts that occur unusually frequently or with high activation. We detect these outliers using the threshold of $\mu_y^{\text{c.s.}} + \alpha^{\text{c.s.}} \times \sigma_y^{\text{c,s}}$ across all context concepts for each class, where we set $\alpha^{\text{c.s.}} = 1$ based on the z-score method, a standard technique for outlier detection [1]. Silhouette score analysis support this choice: Table 5 shows that $\alpha^{\text{c.s.}} = 1$ yields the highest average score (0.67) across datasets, indicating the clearest separation between bias and non-bias context concepts.

---

[9] https://huggingface.co/openai/clip-vit-large-patch14
[10] https://huggingface.co/laion/CLIP-ViT-L-14-laion2B-s32B-b82K
[11] https://huggingface.co/laion/CLIP-ViT-L-14-DataComp.XL-s13B-b90K

Table 5: Average silhouette scores ($\pm$ standard deviation) for different $\alpha$ values across datasets. Each cluster is formed by grouping context and bias concepts.

| Dataset | $\alpha = 0$ | $\alpha = 1$ | $\alpha = 2$ | $\alpha = 3$ |
|---------|--------------|--------------|--------------|--------------|
| Waterbirds | 0.55 | **0.78** | 0.66 | 0.00 |
| CelebA | 0.55 | **0.59** | 0.39 | 0.00 |
| Nico++ | 0.64 | **0.67** | 0.44 | 0.62 |
| ImageNet | 0.62 | **0.71** | 0.61 | 0.27 |
| Food101 | 0.62 | **0.63** | 0.39 | 0.02 |
| SUN397 | 0.61 | **0.65** | 0.42 | 0.05 |
| **AVG** | $0.60 \pm 0.04$ | $\mathbf{0.67 \pm 0.07}$ | $0.49 \pm 0.12$ | $0.16 \pm 0.25$ |

### B.3.3 Additional results: examples of concept categorization

**Example of computing alignment scores.** An illustrative example from the `freight car` class in ImageNet is presented in Fig. 7. Masking out regions activated by the concept `freight train` reduces the cosine similarity relative to the original image (from 0.25 to 0.20). In contrast, retaining only these activated regions increases the similarity (from 0.25 to 0.27). On the other hand, masking regions activated by concepts such as `railroad tracks` or `colorful graffiti art` causes negligible changes in cosine similarity, and retaining only these regions decreases the similarity. Because the alignment score for `freight train` exceeds the threshold, it is classified as a target concept, whereas `railroad tracks` and `colorful graffiti art` fall below the threshold and are classified as context concepts.

**Example of categorized concepts.** Fig. 8 shows the concept activation and their categorization using ConceptScope. The y-axis denotes the concept strengths $f(\mathbf{z})$ computed over all images belonging to class $y$, i.e., $\tilde{f}_{c,y} = \text{avg}_{\mathbf{z}_y \in Z_y}(f(\mathbf{z}_y)_c)$, where $Z_y$ represents the set of image embeddings labeled as class $y$. The x-axis lists concept sorted by type. The right side of the figure shows example images corresponding to each concept type. For the `freight car` class, the primary target concept is a zoomed-in view of a `freight train`, while secondary target concepts include side or front views of `stream locomotives`, as illustrated in the figure. Concepts such as `gravel stones` or `rusted metal chains` are infrequently activated and generally irrelevant. In contrast, `colorful graffiti art` and `railroad tracks` are more commonly activated but are not sufficiently discriminative to independently identify an image as a `freight car`.

**Concepts discovered by ConceptScope provide valuable insights into target classes by capturing both their prototypical representations and diverse visual states.** Fig. 9 illustrates additional examples of categorized concepts from various ImageNet classes, particularly focusing on classes where typical visual representations and their diversity may not be immediately intuitive. ConceptScope clearly identifies prototypical examples through target concepts, reveals variations using context concepts, and highlights dominant patterns as bias concepts.

## C Experiment Details: Concept Prediction

In this section, we provide additional experimental details to supplement §4 of the main paper, including descriptions of datasets, baselines, and further results.

### C.1 Datasets

To assign SAE latents to target attributes, we randomly sample 100 images per class from the training split and select the latent that achieves the highest performance (AUPRC). We then evaluate the selected latents using all test samples.

**Caltech-101 [18].** Caltech-101 dataset is a widely used benchmark for object classification tasks, containing 9,146 images across 102 categories. In our experiments, we use the split provided by HuggingFace [12], which includes 3,066 training images and 6,080 test images.

---

[12]https://huggingface.co/datasets/dpdl-benchmark/caltech101

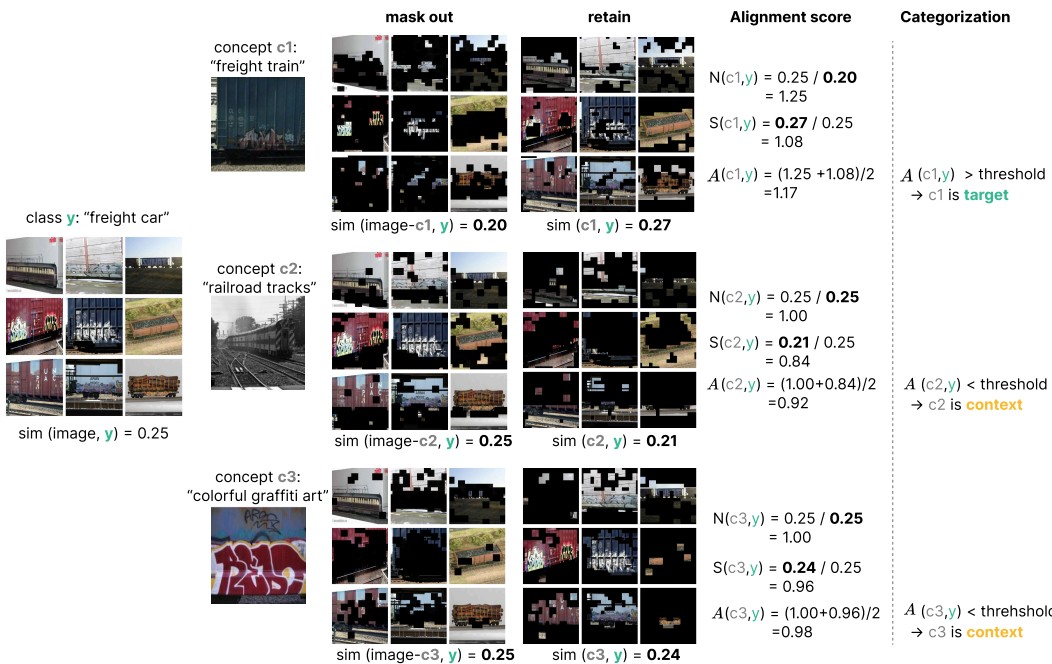

Figure 7: **Examples illustrating the computation of alignment scores.** We present the detailed process for calculating alignment scores of concepts associated with the class `freight car` from the ImageNet dataset, including their actual cosine similarity values, as supplementary to Fig. 2 (c) in the main paper.

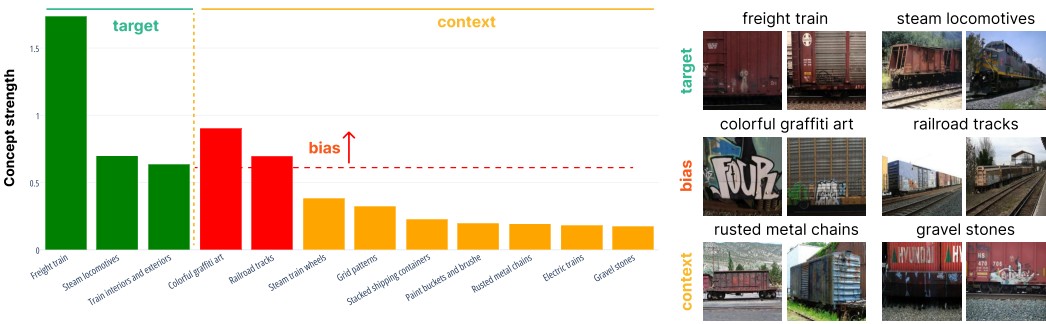

Figure 8: **Examples of concept categorization**. As a supplement to Fig. 2 (d) in the main paper, we show the average concept activation values for each identified concept within the class `freight car` from the ImageNet dataset. Example images corresponding to each concept are provided on the right.

**DTD [9].** The Describable Textures Dataset (DTD) is designed for texture recognition tasks and contains 5,650 images in total, with 3,770 images in the training split and 1,800 images in the test split [13].

**Waterbirds [59].** The Waterbirds dataset contains bird images across four different background types: bamboo forest, forest, ocean, and lake. There are 4,056 images in the training split and 5,794 images in the test split.

**CelebA [44]** The CelebA dataset consists of cropped celebrity images annotated with 40 binary facial attributes. The training split includes 162,770 images, and the test split includes 19,962 images. Although this dataset is widely employed, many attributes are subjective (e.g., `attractive`,

---

[13]https://huggingface.co/datasets/tanganke/dtd

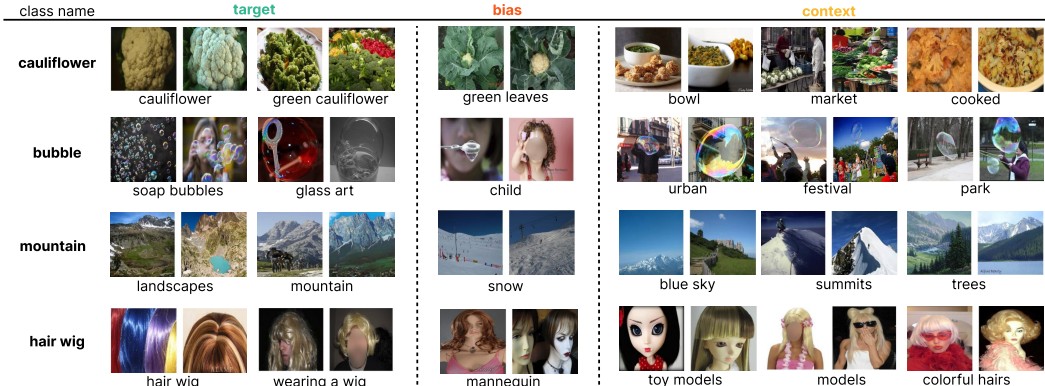

Figure 9: **Additional examples of categorized concepts from ImageNet classes**. To further illustrate the outputs of concept categorization, we provide additional examples of target, context, and bias concepts. ConceptScope effectively identifies prototypical images for target classes, as well as the contexts in which these targets typically appear (context concepts), and highlights the most frequent contexts that may introduce bias (bias concepts).

`big lips`, or `pointy nose`) or overly fine-grained (e.g., `facial hair` divided into `5 o'clock shadow`, `mustache`, `sideburns`, `goatee`). To maintain clarity, we select a subset of attributes primarily related to hairstyles and accessories. The selected 15 attributes are: `bald`, `bangs`, `black hair`, `blond hair`, `brown hair`, `eyeglasses`, `gray hair`, `male`, `smiling`, `wavy hair`, `straight hair`, `wearing earrings`, `wearing hat`, `wearing necklace`, `wearing necktie`, and `young`.

**RAF-DB [36].** The Real-world Affective Faces Database (RAF-DB) is widely used for emotion recognition research and includes seven emotional categories: `happy`, `sad`, `surprise`, `fear`, `disgust`, `anger`, and `neutral`. The dataset contains 12,271 training images and 3,068 test images.

**Stanford40 Actions [72].** The Stanford40 Action dataset contains images of humans performing 40 distinct actions, including `playing guitar`, `riding a bike`, `cooking`, and `reading`. The dataset contains 5,532 training images and 4,000 test images.

## C.2 Baselines

The primary objective of this binary attribute classification experiment in §4 is to verify whether individual latent consistently and selectively correspond to distinct visual concepts. Specifically, a latent dimension representing a given concept (e.g., "chair") should reliably activate across diverse instances of that concept while remaining inactive for unrelated concepts (e.g., "table"). Given the absence of directly comparable baselines, we instead compare our approach against VLM-based methods as suitable alternatives.

**Setup.** For the VLM-based baseline methods used in §4.1 (Table 1) in the main paper, we follow the SSD-LLM [45] framework with modifications. SSD-LLM identifies the distribution of attributes such as backgrounds, actions, or co-occurring objects related to a target class through four steps: (1) caption extraction using VLMs, (2) identification of major attributes from generated captions, (3) iterative refinement of the attribute list, and (4) final assignment of attributes to captions. The original method heavily relies on LLMs to carry out steps 2 to 4, our task focuses on evaluating the presence of a known target attribute. Therefore, we omit steps 2 and 3 (attribute proposal and refinement) and reformulate step 4 as a binary classification task: given a caption, does it imply the presence of the target attribute? This choice follows the growing practice of using LLMs as automatic evaluators or "judges" [22]. Specifically, we use `gemini-2.0-flash-lite` [14] as the LLM, uinsg the following prompt:

---

[14]https://cloud.google.com/vertex-ai/generative-ai/docs/models/gemini/2-0-flash-lite

Prompt

```
''Given the following image caption:  {caption}
Determine whether the attribute {attribute} is present in the caption, either
explicitly or implicitly.  Consider synonyms, paraphrases, and implied
meanings.
If the caption includes the attribute in a semantically meaningful way,
answer 'yes'.  Otherwise, answer 'no'.
Respond strictly with 'yes' or 'no'.''
```

**VLM details.** We employ the BLIP-2 [35] model architecture with a 2.7B parameter model [15]. Input images are resized to $256 \times 256$, and captions are generated using the prompt: "Describe this image in detail." For LLaVA-NeXT [34], we utilize the 8B parameter model [16]. The same prompt used for BLIP-2 is applied, with a maximum token limit set to 256.

## C.3 Additional results: concept prediction

Table 6: **Performance comparison for concept prediction**. We compare our SAE-based method (ConceptScope) against caption-based baselines (BLIP-2, LLaVA-NeXT), reporting average *precision* and *recall* scores across classes.

| Method | Metric | Caltech101 [18] (Objects) | DTD [9] (Textures) | Waterbird [59] (Backgrounds) | CelebA [43] (Facial Attr.) | RAF-DB [36] (Emotions) | Stanford40 [72] (Actions) | Average |
|---|---|---|---|---|---|---|---|---|
| BLIP-2 [35] | *precision* | $0.74_{\pm0.37}$ | $0.50_{\pm0.33}$ | $0.79_{\pm0.18}$ | $0.82_{\pm0.28}$ | $0.37_{\pm0.22}$ | $0.85_{\pm0.16}$ | 0.68 |
|  | *recall* | $0.62_{\pm0.35}$ | $0.39_{\pm0.26}$ | $0.25_{\pm0.09}$ | $0.19_{\pm0.20}$ | $0.19_{\pm0.14}$ | $0.59_{\pm0.21}$ | 0.37 |
| LLaVA-NeXT [34] | *precision* | $0.68_{\pm0.38}$ | $0.35_{\pm0.24}$ | $0.77_{\pm0.20}$ | $0.88_{\pm0.09}$ | $0.57_{\pm0.16}$ | $0.83_{\pm0.21}$ | 0.68 |
|  | *recall* | $0.65_{\pm0.37}$ | $0.63_{\pm0.22}$ | $0.50_{\pm0.14}$ | $0.53_{\pm0.27}$ | $0.43_{\pm0.21}$ | $0.82_{\pm0.18}$ | 0.59 |
| **ConceptScope (Ours)** | *precision* | $0.85_{\pm0.23}$ | $0.62_{\pm0.21}$ | $0.81_{\pm0.09}$ | $0.77_{\pm0.14}$ | $0.55_{\pm0.24}$ | $0.77_{\pm0.16}$ | 0.73 |
|  | *recall* | $0.86_{\pm0.21}$ | $0.57_{\pm0.21}$ | $0.77_{\pm0.06}$ | $0.87_{\pm0.09}$ | $0.58_{\pm0.13}$ | $0.81_{\pm0.12}$ | 0.74 |

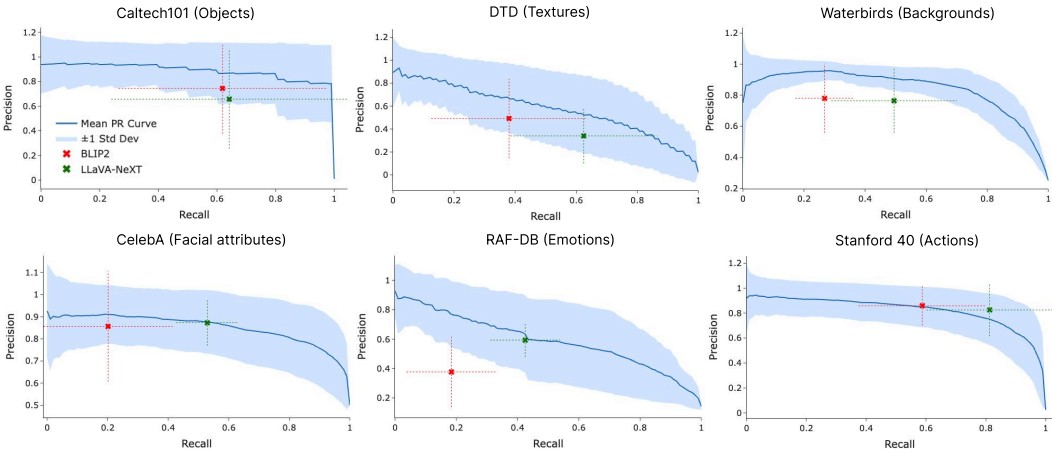

Figure 10: **Average precision-recall curve for attribute prediction using SAE activations.** The shaded blue region represents $\pm1$ standard deviation. Red and green markers indicate the mean precision and recall scores of BLIP-2 [35] and LLaVA-NeXT [34], respectively, with dashed lines showing $\pm1$ standard deviation.

**VLM captions capture general traits but miss specific concepts.** In Table 6, we report precision and recall scores for concept prediction tasks as complementary results to Table 1 of the main paper. Additionally, Fig. 10 shows precision-recall curves for SAE at varying activation thresholds (blue), with baseline performances indicated by red "×" marks for BLIP-2 captions and green "×" marks for LLaVA-NeXT captions. As shown in both the table and the figure, the VLM-based baselines exhibit relatively high precision but significantly lower recall. This reduction in recall primarily arises from

---

[15]https://huggingface.co/Salesforce/blip2-opt-2.7b
[16]https://huggingface.co/llava-hf/llama3-llava-next-8b-hf

Table 7: **Correlation of SAE activation (Ours) & CLIP similarity.** High Pearson and Spearman correlations indicate that SAE activation strength consistently reflects the presence of visual concepts. Each value represents the mean and standard deviation across classes, averaged over four models trained with different random seeds.

| Metric | Caltech101 [18] (Objects) | DTD [9] (Textures) | Waterbird [59] (Backgrounds) | CelebA [43] (Facial Attr.) | RAF-DB [36] (Emotions) | Stanford40 [72] (Actions) | Average |
|---|---|---|---|---|---|---|---|
| Pearson | $0.64 \pm 0.27$ | $0.55 \pm 0.23$ | $0.74 \pm 0.15$ | $0.76 \pm 0.37$ | $0.65 \pm 0.29$ | $0.90 \pm 0.05$ | 0.71 |
| Spearman | $0.45 \pm 0.26$ | $0.53 \pm 0.24$ | $0.91 \pm 0.07$ | $0.74 \pm 0.42$ | $0.60 \pm 0.43$ | $0.71 \pm 0.14$ | 0.65 |

the linguistic diversity inherent in VLM-generated captions. Specifically, as demonstrated in Fig. 11, VLM-generated captions tend to describe general visual characteristics but often fail to capture the specificity needed for identifying target attributes. For example, they describe a `bubbly pattern` merely as `irregular shape`, `blonde hair` as `light-colored hair`, or simply `standing` rather than specifically `fishing`. These observations suggest that while VLMs excel at capturing broad visual descriptions, they are less effective in consistently identifying and quantifying discriminative visual concepts within image datasets.

### C.4 Measuring correlation between SAE activation with CLIP similarity

In §4.1, we report the average correlation between SAE activations and CLIP similarity scores across six datasets. Here, we provide supplementary implementation details and comprehensive results for each individual dataset in Table 7.

**Implementation details.** For each target class, we first select images where the SAE activation for the target concept latent is greater than zero. Next, we compute the cosine similarity between the image embeddings and the text embedding derived from the class label. We sort the selected images based on SAE activation values and partition them into 100 percentile-based groups. For each group, we calculate the average SAE activation and average CLIP similarity. Finally, we evaluate the correlation between SAE activation and CLIP similarity using Pearson and Spearman correlation coefficients.

## D  Experiment Details: Bias Discovery

This section provides additional details for the bias discovery experiments presented in §5.1 and §5.2 of the main paper.

### D.1  Setup

This subsection provides detailed explanations of the experimental setups, evaluation metrics, datasets, and baselines, as well as additional qualitative results, for the bias discovery experiment described in §5.11 of the main paper.

**Evaluation metric.** We measure the performance using Precision@10 [17], which is calculated as follows: for each candidate spurious attribute, the ten test images predicted with the highest confidence for containing that attribute are selected. Precision@10 is then computed as the fraction of these selected images that truly contain the attribute. The final score is the average Precision@10 across all subset–attribute pairs.

### D.2  Datasets

**Waterbirds [59].** The Waterbirds dataset contains two classes: waterbirds and landbirds. The training set comprises 4,795 images. Of these, 95% of the waterbird images have ocean or lake backgrounds, while the remaining 5% feature forest or bamboo forest backgrounds. Conversely, 95% of landbird images have forest or bamboo forest backgrounds, with the remaining 5% featuring ocean or lake backgrounds. The test set consists of 5,794 images, balanced equally across backgrounds (50% water backgrounds, 50% land backgrounds) for both classes.

| Dataset & Class | Input image | SAE activation | BLIP2 | LLaVA-NeXT |
|---|---|---|---|---|
| Caltech101 "ant" | | | "a diagram of a fly and a bee" | "The image is a black and white illustration of a grasshopper. The grasshopper is depicted in a side view, with its body oriented horizontally across the image. The body of the grasshopper is elongated and segmented, with each segment clearly defined." |
| DTD "bubbly" | | | "abstract blue glitter texture background" | "The image displays a close-up of a textured surface with a blue hue. The texture appears to be a pattern of irregular shapes and lines, giving the impression of a cracked or shattered surface. " |
| Waterbrids "bamboo" | | | "a young girl is holding a bird in the air" | "In the image, a young girl is standing in a field of tall, green grass. She is dressed in a blue shirt and a white hat, providing a contrast to the greenery around her. In her hands, she holds a black bird, which is perched on her arm." |
| CelebA "blond hair" | | | "a man with a beard and a black jacket" | "The image is a close-up portrait of a person with a neutral background that appears to be a wall with a patterned texture. The individual has short, light-colored hair that is styled to the side." |
| RAF-DB "angry" | | | "a boy is peeking out of a cardboard box" | "The image shows a close-up of a person's face peeking through a hole in a cardboard box. The person appears to be a young child with fair skin and dark hair. The child's eyes are wide open, and their mouth is slightly open, revealing teeth. The child's expression is one of surprise or shock. |
| Stanford-action "fishing" | | | "a man standing in the grass near a river" | "In the image, a man is standing on a grassy bank, his gaze directed towards a body of water. He is dressed in a blue baseball cap, a plaid shirt, and blue jeans. The water, a light green color, is bordered by a line of trees and shrubs, adding a touch of nature to the scene." |

Figure 11: **Examples of failure cases from VLM-based baselines.** This figure provides illustrative examples explaining why the VLM-based baselines yield lower performance compared to SAE, as reported in Table 1 of the main paper. Although the generated captions accurately describe images at a general level, they frequently lack the specificity needed to effectively capture target attributes, leading to overly simplified or generalized descriptions.

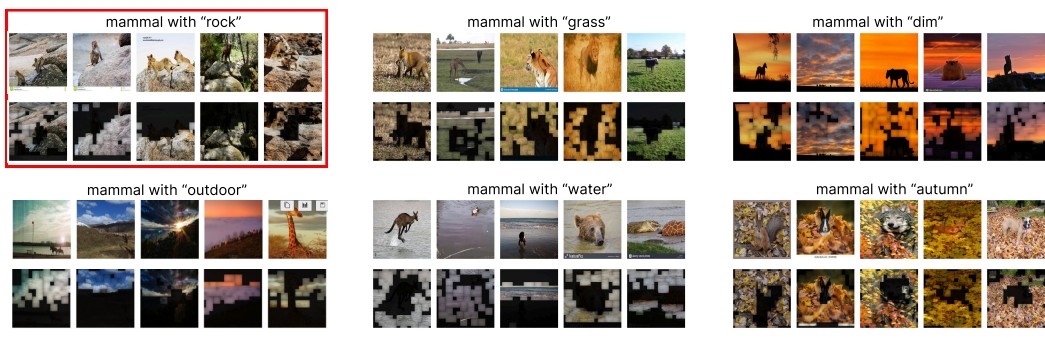

Figure 12: **The qualitative examples of discovered biases.** We illustrate the top 5 test samples of the Nico++ dataset, labeled as "mammals" from each attribute-specific subset. The subset highlighted with a red box corresponds to the attribute correlated with the "mammals" class during training (rock background), while the remaining subsets represent spurious attributes associated with other classes. As shown in the figure, each subset consistently shares the same background attributes, clearly visible in the corresponding segmentation masks shown in the bottom row.

**CelebA [44].** Following prior work, we use CelebA for a binary classification task focused on the attribute blond hair. In the training set, approximately 93% of individuals with blond hair are female. The training dataset consists of 162,770 images, and the test dataset consists of 19,962 images.

**Nico++ [76].** Yenamandra et al. [73] modified the original dataset into a six-way classification task, including the classes: mammals, birds, plants, waterways, landways, and airways. Each class is strongly correlated with a particular background: rocks, grasses, dim lighting, water, outdoor, and autumn, respectively. The severity of the bias in the training dataset is varied across three levels, 75%, 90%, and 95%, indicating the proportion of images in each class exhibiting the correlated background

attribute within the training set. The training sets contain 9,359 samples for NICO++[75], 8,124 for NICO++[90], and 7,844 for NICO++[95]. In contrast, the test set is designed to be balanced: for each class, images are evenly distributed across the six background types. This results in 300 samples per class (50 per background), for a total of 1,800 test images.

## D.3 Baselines

Typical baseline methods for bias discovery involve two stages: first, training models on biased datasets; and second, identifying biases based on model failures in test data through clustering approaches that leverage various features such as model predictions, ground-truth labels, and feature embeddings.

**DOMINO [17].** DOMINO fits Gaussian mixture models (GMMs) using model predictions, ground-truth labels, and CLIP image embeddings for each input image. The GMM clusters data samples based on error types, such as false positives or false negatives. Subsequently, DOMINO generates natural language descriptions of biases by identifying the most similar keywords corresponding to each cluster within the CLIP embedding space.

**FACTS [73].** FACTS extends DOMINO by adding a specialized training objective that amplifies the model's reliance on biased attributes. Specifically, this objective encourages higher confidence scores for samples with biased attributes and lower scores for those without. FACTS then fits separate GMMs within each class to identify distinct subsets associated with bias attributes. Finally, FACTS generates image captions using pretrained captioning models [49] and identifies biases by extracting the most frequently occurring keywords across these captions.

**ViG-Bias [46].** ViG-Bias further improves upon FACTS by incorporating visual explanations into the bias discovery process. It first generates masks using Grad-CAM [61] heatmaps derived from the trained model. In the second stage, ViG-Bias uses these masked image embeddings, rather than original CLIP embeddings, to perform clustering. We use the values reported in the original ViG-Bias [46] paper for comparison in Table 2, and apply the same data split when evaluating our method [17].

**Comparison with baselines.** All baseline methods described above require model training and identify biases based on model errors observed in the test dataset. In contrast, ConceptScope does not require training any additional models and instead directly identifies biases from the dataset itself. Furthermore, baseline methods assume that each class contains only a single type of bias with high correlation (typically greater than 75% of samples per class). However, this assumption rarely holds true for real-world datasets such as ImageNet. Thus, these baseline methods have not been demonstrated to generalize effectively to real-world settings. In contrast, ConceptScope successfully identifies biases directly within real-world datasets, as demonstrated in §5.2.

## D.4 Qualitative results: examples of discovered known biases

As discussed in the last paragraph of §5.1, one key advantage of ConceptScope in identifying biases is its ability to pinpoint specific regions of the input image responsible for introducing bias. Fig. 12 illustrates this capability by presenting examples of biases discovered in the Nico++ dataset. Specifically, the figure shows the top five test images labeled as mammals from each group, selected based on high activation of the latent dimension identified as a bias for each class. Each group consistently exhibits a particular background attribute, as visually highlighted by the segmentation masks provided in the bottom row. Since ConceptScope directly associates latents with clear semantic meanings, it eliminates the need to generate captions to determine the semantic labels of the discovered biases.

## D.5 Additional Results: examples of discovered novel biases

In this subsection, we provide additional examples of discovered biases from real-world datasets, supplementing the findings discussed in §5.2 of the main paper.

**ConceptScope identifies diverse biases, including backgrounds, co-occurring objects, and events, in real-world datasets.** We illustrate examples of biases discovered by ConceptScope in real-world

---

[17]https://github.com/badrmarani/vig-bias-eccv

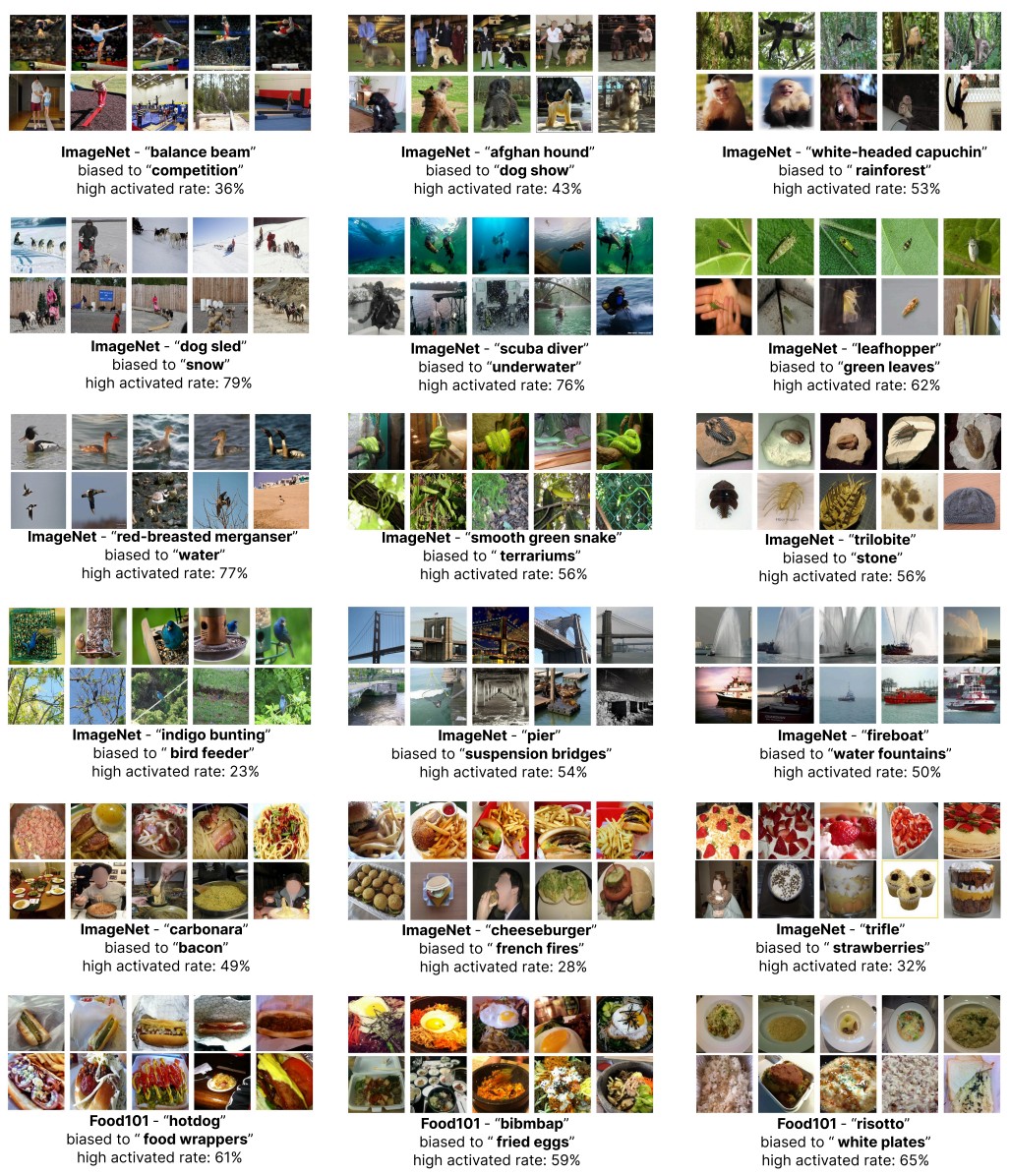

ImageNet - "balance beam"
biased to "competition"
high activated rate: 36%

ImageNet - "afghan hound"
biased to "dog show"
high activated rate: 43%

ImageNet - "white-headed capuchin"
biased to "rainforest"
high activated rate: 53%

ImageNet - "dog sled"
biased to "snow"
high activated rate: 79%

ImageNet - "scuba diver"
biased to "underwater"
high activated rate: 76%

ImageNet - "leafhopper"
biased to "green leaves"
high activated rate: 62%

ImageNet - "red-breasted merganser"
biased to "water"
high activated rate: 77%

ImageNet - "smooth green snake"
biased to " terrariums"
high activated rate: 56%

ImageNet - "trilobite"
biased to "stone"
high activated rate: 56%

ImageNet - "indigo bunting"
biased to " bird feeder"
high activated rate: 23%

ImageNet - "pier"
biased to "suspension bridges"
high activated rate: 54%

ImageNet - "fireboat"
biased to "water fountains"
high activated rate: 50%

ImageNet - "carbonara"
biased to "bacon"
high activated rate: 49%

ImageNet - "cheeseburger"
biased to "french fires"
high activated rate: 28%

ImageNet - "trifle"
biased to "strawberries"
high activated rate: 32%

Food101 - "hotdog"
biased to " food wrappers"
high activated rate: 61%

Food101 - "bibmbap"
biased to "fried eggs"
high activated rate: 59%

Food101 - "risotto"
biased to " white plates"
high activated rate: 65%

Figure 13: **ConceptScope discovers dataset biases in the wild.** We illustrate examples of biases identified by ConceptScope in the ImageNet and Food101 datasets. For each panel, the top row displays samples exhibiting the identified bias attribute, while the bottom row presents samples without the bias attribute. Additionally, we report the proportion of dataset samples with high activation (above a threshold of 0.5) for the corresponding bias latent.

datasets such as ImageNet and Food101 [4]. Our analysis reveals multiple types of biases, including background biases, co-occurring object biases, and event-related biases. Animal-related classes, for example, often exhibit background biases: birds frequently appear near water, reptiles in terrariums, and insects on leaves. Food-related classes commonly display biases involving co-occurring objects, such as burgers paired with french fries. Additionally, we identify biases arising from specific events or contexts, such as balance beams appearing predominantly in gymnastics competitions and certain dog breeds frequently featured at dog shows.

# E   Experiment Details: Model Robustness Analysis

In this section, we provide additional experimental details to supplement §5.3 of the main paper.

## E.1   Full list of models evaluated on ImageNet

The following list includes all models evaluated on the ImageNet and ImageNet-Sketch datasets. The pretrained weights used for evaluation are available at: `https://docs.pytorch.org/vision/stable/models.html`. For models offering multiple pretrained weight options, we consistently select the `ImageNet1K_V1` weights.

1. VGG11 [62]
2. VGG16 [62]
3. VGG19 [62]
4. ResNet18 [23]
5. ResNet34 [23]
6. ResNet50 [23]
7. ResNet101 [23]
8. ResNet152 [23]
9. resnext50 [70]
10. resnext101 [70]
11. ViT-B-16 [14]
12. ViT-B-32 [14]
13. ViT-L-16 [14]
14. ViT-L-32 [14]
15. ConvNeXt-Tiny [42]
16. ConvNeXt-Base [42]
17. ConvNeXt-Small [42]
18. ConvNeXt-Large [42]
19. EfficientNet-B0 [65]
20. EfficientNet-B1 [65]
21. EfficientNet-B2 [65]
22. EfficientNet-B3 [65]
23. EfficientNet-B4 [65]
24. EfficientNet-B5 [65]
25. EfficientNet-B6 [65]
26. EfficientNet-B7 [65]
27. DenseNet121 [25]
28. DenseNet169 [25]
29. DenseNet201 [25]
30. WideResNet50 [74]
31. WideResNet101 [74]
32. SwinTransformer-Base [40]
33. SwinTransformer-Tiny [40]
34. SwinTransformer-Small [40]

