# OpenReview forum: "ConceptScope: Characterizing Dataset Bias via Disentangled Visual Concepts"
_NeurIPS.cc/2025/Conference — NeurIPS 2025 poster_

### Official Review · Reviewer_usgc · 2025-06-26

**Clarity:** 3
**Significance:** 3
**Originality:** 3
**Rating:** 4
**Confidence:** 3

**Summary:**

This paper introduces ConceptScope to detect biases in datasets by disentangling visual concepts through a sparse autoencoder. This is achieved by training a linear sparse autoencoder to reconstruct visual features for a pretrained model. The authors found that the activation of sparse codes reveals one concept, e.g., object and texture. Based on this, the normalized alignment score is proposed to distinguish concepts for target, contextual, and bias. The approach is able to discover biases in any datasets and achieves good performance in 5 benchmarks for bias discovery. ConceptScope can be also used for robustness evaluation.

**Questions:**

What are the advantages of SAE compared to clustering in concepts discovering?

How Figure 5 diagnoses model robustness? It is reasonable to find the target with biases getting the best accuracy. The trend cannot tell me how robust the model is. What is the robustest model from 34 models?

**Ethical Concerns:**

["NO or VERY MINOR ethics concerns only"]

**Final Justification:**

This work proposes ConceptScope to detect the biases on the data, which is novel to provide a model robustness evaluation without OOD benchmarks. The response from authors resolves most of my concerns. However, we can see from the response that there is a gap between the proposed concept discovery and the SOTA. Since it is not the major contribution and can be improved in the future work, this work has the potential to be applied for model robustness diagnoses.

**Limitations:**

yes

**Quality:**

3

**Strengths And Weaknesses:**

Strength:
1. The proposed method for concept discovering via SAE is simple yet effective, providing insights on diagnosing biases in datasets.
2. The metrics of detecting bias is novel. The experimental results reveal good performance in detecting biases among datasets.
3. The paper is easy to follow.

Weakness:
1. This work only validates the concept discovering for CLIP. As we know SAM is a powerful segmentation model, how well it will be to apply SAE into SAM?
2. The comparison to feature clustering is missing. The features can generate good segmentation mask for concepts if proper threshold is used [1]. Therefore, performing clustering can also get visual concepts. It is vital to know the advantages of sparse autoencoder comparing to clustering.

Language Issue:
1. In line 217, "For each target attribute y, we image-level activation value..."

[1] Wu J, Mo S, Awais M, et al. Masked momentum contrastive learning for zero-shot semantic understanding[J]. arXiv preprint arXiv:2308.11448, 2023.

---

> ### Author Rebuttal · Authors · 2025-07-31
>
> ***Thank you for the constructive feedback. We hope our responses address the reviewer's concerns.***
>
> ### *Summary of response*
> - (W1) Regarding the concern that SAE is only applied to CLIP, we clarify that our goal is to **categorize concepts and analyze class-level biases and visual characteristics**. We also provide a quantitative comparison across different CLIP variants for the concept prediction task.
> - (W2, Q1) Regarding the lack of comparison with feature clustering methods, we note that this line of work is fundamentally different. Instead, we **qualitatively compare ConceptScope with SAM-based concept discovery** and show that our method captures more fine-grained object and texture concepts that baseline methods miss.
> - (Q2) Regarding questions about model robustness diagnosis, we explain that **ConceptScope enables practical and interpretable robustness assessment** through concept-based subgroup analysis, even in the absence of explicit distribution shift datasets.
>
> ---
> ### *(W1) Discussions regarding SAM-based concept analysis*
> While our work applies SAE to the CLIP encoder, a similar approach can be applied to other vision models, such as Segment Anything Model (SAM). As mentioned in our related work (line 92), SAE-based concept analyses have been applied to diffusion models, suggesting broader generalizability.
>
> Applying SAE to the encoder of SAM and comparing its concepts with those from CLIP is indeed a promising direction for future research. However, this is beyond the current scope of our work. Here, our primary goal is to **categorize decomposed concepts** of CLIP representation **into target, bias, and context types** to **analyze class-level biases and visual characteristics** in class-labeled datasets.
>
> Although comparing SAE across model architectures is outside the scope of this work, we evaluate its **robustness across different CLIP variants**, (all ViT-L/14) pretrained on different datasets:
>
> - OpenAI CLIP (used in our main experiments)
> - DataComp CLIP
> - LAION CLIP
>
> To evaluate performance, we conduct a concept prediction task (Tab. 1, Sec. 4.1 of the paper) across six benchmark datasets (see Table 1 of the paper).
>
> | Model|Caltech101|DTD|Waterbird|CelebA|RAF-DB|Stanford40|AVG |
> |-|-|-|-|-|-|-|-|
> |BLIP2*|0.74|0.34|0.60|0.40|0.25|0.47|0.41|
> |LLaVA-NeXT*|0.64|0.42|0.62|0.59|0.47|0.43|0.47|
> |OpenAI (**ours**)|0.83|0.52 |0.78|0.67|0.55|0.75|0.68|
> |DataComp (**ours**)|0.85| 0.52|0.76|0.68|0.39 |0.68|0.65|
> | LAION (**ours**)| 0.80  |  0.63 | 0.78| 0.69| 0.44|0.64|0.66|
>
> *: baseline methods from Tab. 1
>
> Despite some variation across CLIP variants, our method consistently outperforms baseline models (0.41, 0.47) by a substantial margin, achieving average F1 scores between 0.65 and 0.68.
>
>
> It is also worth mentioning that, to the best of our knowledge, **no prior work has quantitatively evaluated both the concept prediction task (Section 4.1) and their corresponding spatial attributions (Section 4.2)** of SAE concepts. Evaluating these aspects systematically is one of our key contributions.
>
> ---
> ### *(W2, Q1) Comparison to feature clustering*
> The paper referenced by the reviewer primarily focuses on enhancing zero-shot segmentation using visual prompts (e.g., point-click) for self-supervised segmentation models. This direction is fundamentally different from ours. In contrast, our goal is to **automatically discover semantically meaningful and class-dependent visual concepts**, without any human prompts or pre-defined labels, and use them to **analyze dataset-level bias**.
>
> Unlike traditional segmentation tasks, producing a high-quality mask is not the main purpose of segmentation derived from SAE spatial attribution in our framework. Instead, these attributions are used when computing the alignment score (Eq.(3) and Eq.(4)), to ablate or isolate each concept from an image. Our focus is to analyze concept–class relationships and **categorizing concepts** (target, context, bias), not on optimizing the segmentation quality.
>
> A relevant comparison candidate is HU-MCD [1], which combines SAM-generated region masks with feature clustering to extract object- or background-level concepts. Although HU-MCD demonstrates potential in discovering high-level visual entities (e.g., object parts, background elements), it has notable limitations:
> - It struggles to identify fine-grained attributes such as textures, colors, or object subtypes.
> - It provides no formal quantification of how each discovered concept relates to class identity or dataset bias.
>
> A direct quantitative comparison would ideally require a benchmark dataset annotated with diverse concept types (e.g., objects, textures, colors, contexts). However, such a benchmark does not currently exist, to the best of our knowledge. Alternatively, we provide **qualitative comparisons** using the ImageNet classes—“Airplane,” “Beach Wagon,” and “Tailed Frog”—selected from Figure 3 of the HU-MCD paper.
>
> We observed that
> - HU-MCD identifies broad object or background concepts like “airplane,” “airplane tail,” and “sky". In contrast, SAE discovers **more fine-grained and semantically diverse concepts**, such as "commercial airplanes", "military aircraft", and "jet" as well as different states such as "landing" and "flying".
> - For the class “Tailed Frog,” SAE captures **distinct attributes** such as "dotted skin texture", "frog eyes", along with diverse backgrounds such as "mud", "terrariums", "walls", "swamp".
> - In the case of “Beach Wagon,” SAE reveals **detailed components**, including "car grille", "car bumper", "backlights". Additional car-related concepts are provided in Supplementary Figure 1 (“seat belts,” “side mirrors,” “car accidents”).
>
> These findings highlight SAE’s strength in producing nuanced, human-interpretable, and semantically disentangled concepts that go beyond what region-based clustering typically provides. Moreover, ConceptScope goes beyond discovering concepts—it also reveals which of these concepts appear frequently and disproportionately in specific classes, allowing us to identify potential sources of dataset-induced bias. To the best of our knowledge, no prior work provides this level of insight into how visual concepts are distributed across training data.
>
> [1] Grobrügge, Arne, et al. "Towards Human-Understandable Multi-Dimensional Concept Discovery." CVPR 25.
>
> ---
> ### *(Q2) Model robustness diagnoses*
> *We will add the clarification for Section 5.3 in the camera-ready version.*
> Model robustness is typically evaluated by measuring the performance drop on distribution-shifted test sets—such as from ImageNet to ImageNet-v2 or ImageNet-Sketch. However, in many domains (e.g., food or scene classification), such curated out-of-distribution (OOD) benchmarks are unavailable, and constructing them manually can be both costly and time-consuming.
>
> ---
> **Concept-based distribution shifts**
>
> **ConceptScope** provides a complementary approach that enables robustness evaluation **within the original test set**, even when no distribution-shifted dataset is available. To do this, we create four subgroups of test samples with varying concept distributions.
>
> As shown in Figure 5, for the class *“hippo”*, ConceptScope identifies **“hippo head”** as a target concept and **“water”** and **“zoo”** as bias concepts. We then define the following subgroups:
> 1. **Group 1**: Images containing both the hippo head and water and/or zoo.
> 2. **Group 2**: Images with the hippo head, but without water or zoo—i.e., a different background context.
> 3. **Group 3**: Images featuring water and/or zoo, but where the hippo head is either absent, unclear, or too small.
> 4. **Group 4**: Images labeled as “hippo” but lacking both the hippo head and the identified bias concepts (water/zoo).
>
> These groups can be interpreted as forming a spectrum of distribution shift. **Group 1** represents the most frequent and typical setting seen during training, where both the target and bias concepts co-occur. In contrast, **Group 4** represents a more unusual or outlier scenario, where neither the target nor the bias concepts are present—posing a stronger generalization challenge for the model.
>
> This setup allows us to simulate concept-based distribution shifts and assess model robustness without requiring a separate OOD dataset.
>
> ---
> **Interpreting subgroup accuracy**
>
> In Fig. 5(b), the y-axis represents subgroup accuracy (with the four groups shown in different colors), and the x-axis indicates the average accuracy across all four subgroups. Each vertical line corresponds to a single model, with four colored dots representing that model’s accuracy on each subgroup.
>
> Among the 34 models evaluated, **ConvNeXt-Large** achieves the highest average accuracy and is shown on the rightmost line (in both ImageNet (IN) and ImageNet-Sketch (IN-S)). Notably, it also performs best on **Group 4**, the most challenging subgroup. We observe that models with higher accuracy on **Group 1** (the most frequent training setting) also tend to perform better on **Group 4** (the rare or outlier setting).
>
> This trend is consistent with prior findings [1], which show that out-of-distribution (OOD) performance often correlates with in-distribution performance. Crucially, our approach enables such robustness analysis **without requiring an external OOD test set**, using only concept-aware subgroups derived from the original data.
>
> To clarify,  our goal in this experiment is not to rank all 34 models by robustness, but rather to demonstrate that ConceptScope can analyze how and why models fail. This capability makes ConceptScope a useful tool for estimating or understanding model robustness, especially in cases where no special OOD test set exists.
>
> [1] Miller, John P., et al. "Accuracy on the line: on the strong correlation between out-of-distribution and in-distribution generalization." ICLML, 21.
>
> ---
> ### *Typo*
> Thanks for pointing out, we will revise accordingly.

---

> > ### Comment · Reviewer_usgc · 2025-08-04
> >
> > Thank you for your feedback. The purpose of mentioning SAM and feature clustering methods is to see the quality of segmentation masks. While segmentation is not the main contribution, it is a prerequisite for SAE to extract concepts from images. Binary classification indicates the existence of concepts, but it is unable to show that SAE find the correct concepts. For example, LLaVa-NEXT finds the concept for water but recognises the cloud as water. I am curious about the reliability of Sparse Autoencoders in Reliable Concept Extractors. Therefore, I suggested zero-shot segmentation algorithm to compare the reliability in extracting the concept regions.
> >
> > Regarding to model robustness, the authors solved my concerns.

---

> ### Author Response · Authors · 2025-08-05
>
> Thank you for the reply — we’d be happy to continue the discussion and respond:
>
> ### **Semantic attribution**
>
> We would like to clarify that the SAE model in our paper **is capable of producing segmentation masks** for spatially attributing these concepts (Fig. 3). These masks are coarse-grained (at the token level), which is in contrast to the dense, per-pixel masks typically produced by conventional semantic segmentation models.
>
> In Fig. 3, we already validated our model against relevant baselines (LLaVA-Next and BLIP2). As we validate in Fig. 3, the spatial attribution outperforms relevant baseline models.
>
> Several other methods [1,2,3] have been proposed for open-vocabulary segmentation to overcome the limitations of traditional segmentation methods, which can only segment a fixed set of categories. These approaches enable zero-shot segmentation using category names, without requiring dense pixel-level annotations during training.
>
> - **MaskCLIP** [1] uses a class-agnostic mask proposal network (e.g., Mask R-CNN), followed by matching predicted regions with CLIP text embeddings for classification.
> - **ODISE** [2] employs a diffusion model to enhance mask quality.
> - **EBSEG** [3] fuses image features from the **SAM encoder** with CLIP embeddings to better capture spatial information.
>
> We evaluate SAE using *mean binary IoU across 150 classes* (*) of ADE20K, treating each class as an independent binary segmentation task. In contrast, the baseline methods--MaskCLIP, ODISE, and EBSEG--report *multi-class mIoU*, where all categories are predicted jointly. Although these evaluation protocols differ, they give some guidance that the segmentation performance of SAE ends up in the same ballpark as these specialized open-vocabulary segmentation methods:
>
> | Method                 | IoU (%) |
> | ---------------------- | ------- |
> | **SAE (Ours)**             | 28.4*   |
> | [1] MaskCLIP (Table 2) | 23.7    |
> | [2] ODISE (Table 1)    | 28.7    |
> | [3] EBSEG (Table 1)    | 32.8    |
>
> ### **References**
>
> [1] Open-Vocabulary Universal Image Segmentation with MaskCLIP — Ding, Wang, Tu. ICML 2023.
>
> [2] Open-Vocabulary Panoptic Segmentation with Text-to-Image Diffusion Models (ODISE) — Xu et al. CVPR 2023.
>
> [3] Open-vocabulary semantic segmentation with image embedding balancing.— Shan, Xiangheng, et al. CVPR 2024

---

> ### Author Response · Authors · 2025-08-07
>
> As we approach the end of the discussion phase, we would like to kindly remind you that we have added further clarification, which we hope addresses your remaining concerns. If you have any additional questions or suggestions, we would be happy to discuss them.

---

### Official Review · Reviewer_6bLm · 2025-06-29

**Clarity:** 2
**Significance:** 2
**Originality:** 2
**Rating:** 5
**Confidence:** 3

**Summary:**

The paper leverages sparse auto-encoders (Cunningham et al [7]) to discover concepts related to the image target class, the context, and potential bias. Methodologically, it follows a similar line work from Lim et al [12], which are leveraging sparse autoencoders to reveal selective remapping of concepts. The paper relies on multimodal LLM to discover concepts that emerge from a vision encoder trained on ImageNet. Evaluation assesses whether such framework can predict concepts in various datasets and discover biases present in visual datasets.

**Questions:**

I am happy to increase my score with improvements regarding discussion and comparison with missing related work, details about how concepts are annotated and their limitations.

**Ethical Concerns:**

["NO or VERY MINOR ethics concerns only"]

**Final Justification:**

The rebuttal has addressed my concerns and is clarifying aspects regarding the related work, the annotation process, and properties of the method.

**Limitations:**

Yes.

**Paper Formatting Concerns:**

N.A.

**Quality:**

2

**Strengths And Weaknesses:**

**Strengths**

*S1. The method enables to discover novel biases*
* Figure 4 illustrates novel biases being discovered by the proposed method. It is valuable that the method goes beyond known biases in a dataset and enables to discover relations between concepts that can affect the object classification performance.

*S2. Concepts are visually grounded*
* Figure 3 illustrates how discovered concepts are grounded in images. This offers a better interpretability of the discovered concepts.

**Weaknesses**

*W1. Related work section is insufficient*
* The paper ignores related works in bias discovery, which is a new but already rich field. It would be beneficial to discuss differences in terms of datasets, biases, and evaluation protocol. For example, here are a few:
  * Beery et al. Recognition in Terra Incognita. ECCV 2018.
  * Hirota et al. SANER: Annotation-free Societal Attribute Neutralizer for Debiasing CLIP. ICLR 2025.
  * Lang et al. Explaining in Style: Training a GAN to explain a classifier in StyleSpace. ICCV 21.
  * Li et al. Discover the Unknown Biased Attribute of an Image Classifier. ICCV21.
  * Zhang et al. Discover and Mitigate Multiple Biased Subgroups in Image Classifiers. CVPR 24.

*W2. The annotation process for concepts is unclear*
* The paper only mentions that it "annotates latent dimensions using multimodal LLM based on reference with localized activations". How many concepts were annotated, and how was this decided? Are there any manual checks to verify the validity of the annotations? Can any bias be introduced in the annotation process? Is there any manual input in the annotation process, or all is done by an LLM? In the current form, it is hard to understand how concepts are discovered and reproduce the annotation process.

*W3. Out-of-distribution concepts are unclear*
* The paper is based on discovering concepts present in ImageNet. What if we want to characterize a dataset for which concepts were not discovered?
* Furthermore, what is a sufficient set of concepts to be able to perform the different tasks assessed in the paper?
* On a related topic, in Table 1, the performance on the DTD dataset is quite poor. What makes this dataset challenging? Similar question for the RAF-DB dataset.

---

> ### Author Rebuttal · Authors · 2025-07-31
>
> ***Thank you for the constructive feedback. We hope our responses address the reviewer's concerns.***
>
> ### *Summary of responses*
> - (W1) We clarify that unlike prior work focused on model bias, **ConceptScope directly analyzes the training dataset**, offering a fundamentally different approach.
> - (W2) We explain that **concept extraction relies solely on SAE**, not on multimodal LLMs. LLMs are used only post hoc for interpretation and do **not affect the core algorithm**.
> - (W3) We show that **SAE-learned concepts generalize across datasets**, supported by empirical activation statistics.
>
> ---
> ### *Clarification on summary*
> Our proposed method does not rely on multimodal LLM to discover concepts. We use SAE for concept extraction. We clarified the purpose and role of generated annotation in (W2) Clarifying the concept annotation process.
>
> ---
> ### *(W1) ConceptScope analyzes dataset*
> While existing work addresses important aspects of model bias, **none of them directly analyze the structure of the training dataset**. In contrast, our method focuses on **automatically discovering and characterizing bias by analyzing visual concept distributions in the training data**, which is largely orthogonal to these prior efforts. Below, we outline the key distinctions:
>
> **Model robustness benchmarks**
> - **Beery et al. (ECCV 2018)** introduced a benchmark to evaluate model robustness under distribution shifts across geographical locations. While it highlights the importance of dataset-induced bias, it does not propose methods for identifying or interpreting biased attributes within the dataset itself.
>
> **Societal bias mitigation**
> - **Hirota et al. (ICLR 2025, SANER)** aim to mitigate societal biases (e.g., gender, race) in CLIP by neutralizing sensitive attributes. In contrast, our work does not assume any predefined sensitive attributes. Instead, we focus on **automatically discovering unknown or subtle biases** directly from the dataset.
>
> **Concept-based explanation methods**
> - **Lang et al. (ICCV 2021, StylEx)** and **Li et al. (ICCV 2021)** use generative models (e.g., StyleGAN2) to discover interpretable latent directions that explain classifier decisions. While ConceptScope also extracts interpretable visual concepts—using a sparse autoencoder over frozen CLIP features—our goal is fundamentally different: we use these concepts to **characterize the dataset itself**, not to explain predictions of a specific model.
>
> **Subgroup discovery via failure analysis**
> - **Zhang et al. (CVPR 2024)** identify model bias by segmenting test samples into subgroups with poor performance and finding shared visual features. This is related to baseline methods we compare against in Table 2. However, these methods detect dataset bias indirectly—through downstream model behavior. If the training dataset is biased (e.g., birds frequently co-occurring with ocean backgrounds), but the test set lacks bias-conflicting examples (e.g., birds without ocean), such methods cannot detect the presence of bias in the training data or the model’s reliance on it.
> - More importantly, as discussed in the main paper (lines 80–81) and Supplementary Section A.5 (lines 251–258), these methods do not directly inspect the training data and thus cannot reveal **which training samples contain bias-inducing features** or **how prevalent** those features are.
>
> In summary, **ConceptScope directly analyzes the training dataset** to uncover the distribution of learned concepts, including stereotypical visual states, contextual co-occurrences, and potential sources of spurious correlations (see **Supplementary Figures 3 and 4**). This enables a more comprehensive and interpretable understanding of dataset-level bias, independent of any downstream model or task. We believe this ability to inspect and interpret the internal structure of the training data itself represents a significant and novel contribution to the field of dataset bias analysis.
>
> ---
> ### *(W2) Clarifying the concept annotation process*
> **Purpose and role of VLM annotation**
>
> The use of a VLM for concept annotation is a post-hoc step intended to make the extracted concepts easier for human interpretation. Importantly, this step does **not directly influence** the concept categorization, which includes alignment score or concept strength computation and thresholding.
>
> **Detailed annotation procedure**
>
> We detail the annotation procedure—including the prompt template—in **Supplementary Section A.2**, with examples of concepts and their annotated labels shown in **Supplementary Figure 1**. We first collect the top-activating image patches for each concept and extract their corresponding segmentation masks, following the standard procedure introduced in PatchSAE [28]. We then apply a GPT-4o to generate short natural language descriptions from these visual inputs. The annotation process is fully automated, except for light prompt tuning: we tested several prompt formats and selected one that consistently yielded concise, accurate, and semantically rich descriptions.
>
> **Annotation quality**
>
> Although we do not manually verify every annotation, qualitative inspection shows that the generated descriptions are consistent and meaningful across similar visual patterns (see **Supplementary Fig. 1** and **Fig. 4**). Examples include concise, interpretable phrases like "knitted fabric" or "yellow objects."
>
> To assess consistency, we compared generated captions by GPT-4o and LLaVA-NeXT (`llama3-llava-next-8b`). The average cosine similarity was **0.754** (±0.119) in CLIP’s text embedding space and **0.862** (±0.050) in OpenAI’s `text-embedding-ada-002` space, indicating strong agreement and convergence toward simple, intuitive descriptions. Examples of caption pairs:
> - LLaVA-NeXT vs. GPT-4o generated -- CLIP similarity, OpenAI similarity
> - curtains vs. curtains -- 1.00, 1.00
> - rooster vs. red rooster -- 0.81, 0.93
> - vehicles vs. convertible cars -- 0.76, 0.88
> - textured vs. knitted fringe -- 0.53, 0.81
>
> We acknowledge that annotation quality may vary across different VLMs and consider alignment with human perception an important area for future work.
>
> ---
> ### *(W3) Generalizability of ConceptScope*
> **Concepts beyond ImageNet**
>
> Although the SAE is trained on ImageNet, the resulting visual concepts are not limited to the ImageNet dataset; rather, they **generalize well to other datasets**--commonly used classification dataset for CLIP zero shot evaluation--such as Food101 and SUN397 (scene) (Lim et al., 2025). This generalization is expected, as the SAE is not trained using ImageNet class labels. Instead, it learns to reconstruct image representations from a general-purpose foundation model—the CLIP vision encoder—which is assumed to have already captured diverse visual concepts, as evidenced by its strong zero-shot performance.
>
> If we aim to characterize a dataset that is significantly different from ImageNet—such as medical images—we should train SAE on that specific domain and/or with a more suitable backbone model. As demonstrated in prior work (CytoSAE [1], Mammo-SAE [2]), applying SAE to medical images is feasible, and a similar pipeline can be adopted for domain-specific analysis.
>
> [1] Dasdelen et al. (2025). CytoSAE: Interpretable Cell Embeddings for Hematology. arXiv.
>
> [2] Nakka et al. (2025). Mammo-SAE: Interpreting Breast Cancer Concept Learning with Sparse Autoencoders. arXiv.
>
> **Sufficient set of concepts**
>
> We have not yet systematically analyzed how many or how granular concepts are sufficient for different tasks, but we consider this an important direction for future work.
>
> Although we do not provide a standardized benchmark through extensive comparisons, we present **empirical evidence** suggesting that approximately 3,000 concepts are sufficient for meaningful concept extraction and categorization.
>
> In this work, we found that the SAE learns around `3,000` meaningful concepts. Among these, `2,537` concepts are frequently activated in ImageNet, `259` in Food101, `1328` in SUN397, and so on. On average, approximately `20` concepts are activated per class.
>
> **Performance on fine-grained dataset**
>
> Since our concepts are derived from CLIP’s representations, they are inherently constrained by CLIP’s learned knowledge. However, using vision foundation models fine-tuned on domain-specific datasets like RAF-DB or DTD could improve performance by yielding concrete and task-relevant latent concepts.
>
> RAF-DB is a facial expression recognition dataset with 7 emotion categories. It is particularly challenging because some emotions—such as "fear" and "surprise"—are difficult to distinguish even for humans due to their subtle visual differences. Although CLIP’s large-scale training data may contain some facial images with emotional expressions, it was not explicitly trained on facial expression recognition tasks. As a result, it lacks the fine-grained supervision needed to capture subtle facial cues, leading to relatively low zero-shot performance on RAF-DB (39.8%) [1].
>
> DTD is a texture classification dataset with 47 fine-grained texture categories. Many of these textures are abstract and visually similar, lacking distinctive semantic features. This makes zero-shot classification difficult, and CLIP achieves an accuracy of 50.4% on DTD [2].
>
> [1] Zhao, Zengqun, et al. "Enhancing zero-shot facial expression recognition by llm knowledge transfer." WACV 2025.
>
> [2] Wu et al. "How well does CLIP understand texture?." arXiv(2022).

---

> > ### Author Response · Authors · 2025-08-05
> > **Gentle reminder for discussion**
> >
> > Dear Reviewer,
> >
> > Thanks again for your thoughtful review. We addressed your questions and would appreciate if you could re-evaluate our contribution in the light of our rebuttal. If concerns are remaining, we are happy to discuss further during the discussion period.

---

> > > ### Comment · Reviewer_6bLm · 2025-08-06
> > >
> > > Thank you for the rebuttal, it addresses my concerns.
> > >
> > > It would be helpful to know how the rebuttal will be integrated in the paper.

---

> > > > ### Author Response · Authors · 2025-08-06
> > > >
> > > > ***We sincerely appreciate the reviewer for reading our rebuttal and the positive assessment.***
> > > >
> > > > We will integrate the clarifications from the rebuttal in the main paper as follows:
> > > >
> > > > > (W1) ConceptScope analyzes dataset
> > > > - We will expand the **Section 2. Related work - Bias identification** (line 77-85) to include the additional related works discussed in the rebuttal. Specifically, we will emphasize that the key distinction of our method lies in directly analyzing the structure of the training dataset, rather than only focusing on model behavior.
> > > >
> > > > > (W2) Clarifying the concept annotation process
> > > > - We will revise **Section 3.2 - Interpreting learned concepts** (line 156-166) to incorporate the annotation process originally detailed in the supplementary material. Specifically, we will clarify that this process does not influence the concept categorization algorithm.
> > > > - Additionally, we will provide qualitative examples of concept annotations, corresponding images, and consistency analysis at the end of **Section 4 - Sparse Autoencoders are Reliable Concept Extractors**. More examples and experimental details will be included in **Supplementary Section A.2**.
> > > >
> > > > > (W3) Generalizability of ConceptScope
> > > > - At the end of **Section 3.2 - Construction the Concept Dictionary**, we will emphasize that SAE concepts learned from ImageNet generalize well to other datasets.
> > > > - In **Section 6 – Conclusion and Limitations**, we will discuss how ConceptScope can be extended to domains that differ significantly from ImageNet (e.g., medical imaging; citing CytoSAE and Mammo-SAE) by training SAEs on domain-specific foundation models, highlighting the broader impact of our work and its potential to inspire future research.
> > > > - We will also include the number of highly activated concepts per dataset in **Section 5 -ConceptScope for Dataset and Model Bias Analysis**.
> > > > - Lastly, we will clarify in **Section 4.1 – Evaluating Concept Prediction Performance** that the lower performance observed on RAF-DB and DTD (Table 1) is due to the inherent challenges of these datasets and limitations of CLIP in those domains.
> > > >
> > > > Thank you again for your insightful feedback, which has helped us improve the clarity and completeness of the paper.

---

> > > > > ### Comment · Reviewer_6bLm · 2025-08-06
> > > > >
> > > > > Thank you for clarifying and explaining how the paper will be improved. I will increase my score.

---

> ### Author Response · Authors · 2025-08-07
>
> We appreciate your constructive feedback once again and are deeply grateful for your positive assessment and for the increase in the score. We will carefully revise the manuscript as outlined above.

---

### Official Review · Reviewer_QZaG · 2025-07-01

**Clarity:** 2
**Significance:** 3
**Originality:** 3
**Rating:** 4
**Confidence:** 4

**Summary:**

This paper introduces ConceptScope, a novel framework for identifying and characterizing biases within image classification datasets. The approach leverages a Sparse Autoencoder (SAE) trained on visual encoder representations (e.g., from CLIP) to automatically discover a dictionary of disentangled visual concepts. The framework then categorizes these concepts for each class as "target" (essential to the class definition), "contextual" (correlated but not essential), or "bias" (contextual concepts that spuriously co-occur with high frequency). The authors empirically validated the approach by investigating its effectiveness in concept prediction and localization tasks in comparison to VLM-based baselines. ConceptScore was then compared to other bias detection approaches in tasks such as detecting known biases in benchmarks like CelebA and Waterbirds. Its capacity to discover novel, unannotated biases in datasets like ImageNet was also investigated. Finally, the paper demonstrates a practical application for diagnosing model robustness by partitioning test sets based on concept activation strength.

**Questions:**

- Clarification on Definitions: Could you provide a more precise definition of "concept strength"? What is the exact threshold used to determine if a strength is "high" when identifying a bias concept? The paper states it is $\mu_{c', y}​+\sigma_{c', y}$​​ , which seems to imply that there will always be concepts identified as biased. Is this intended?
- Handling of Multiple Biases: Is the framework fundamentally limited to identifying only one bias concept per class? How would it perform in a scenario with several distinct, equally prominent spurious concepts?
- Supervision and Manual Effort: Could you clarify which steps of the ConceptScope pipeline are fully automated and which require manual intervention? Specifically, what is the exact process and VLM-prompting strategy used to annotate the latent concepts in the dictionary, and how robust is this to the choice of VLM?
- Experimental Details:
  - What population are the error bars in Figure 3 calculated over?
  - Could you provide a brief definition or a more direct reference for the Precision@10 metric used for the bias discovery task? The paper cites a source but does not define the metric itself.
  - Computational Cost: How does the computational cost of ConceptScope (including SAE training) compare to the baseline methods in Table 2, which require training models on biased datasets to identify failure modes?
  - Validity of Target/Context Separation: The stratification of concepts into "target" and "contextual" seems critical. How does the framework ensure this initial step is not itself biased? For example, using CLIP similarity to measure necessity and sufficiency might inherit any biases present in CLIP's text-image embedding space.

**Ethical Concerns:**

["NO or VERY MINOR ethics concerns only"]

**Final Justification:**

The authors have addressed my concerns during the rebuttal. I would be more confident in recommending accepting the paper if I could read an updated version of the manuscript, but as the authors provided a clear plan for how they are going to incorporate the changes, my final score is "Borderline Accept".

**Limitations:**

I think it is important to include in the limitations that the core assumption that concepts can be cleanly separated into "target-essential" and "contextual" categories is a strong one. As raised in the weaknesses, this process may introduce its own biases by enforcing a narrow, monolithic definition of a class (e.g., "happy face must have a smile"), potentially causing the framework to mischaracterize the dataset's diversity.

**Paper Formatting Concerns:**

My major concerns are regarding the clarity of the manuscript, especially regarding the questions raised in the section above.

**Quality:**

3

**Strengths And Weaknesses:**

Strengths:
- Discovery of Novel Biases: A significant strength is the framework's demonstrated ability to uncover previously unannotated biases in widely used datasets, such as cultural biases in the "bridegroom" class or object co-occurrence biases in the "necklace" class in ImageNet. This highlights the practical utility of the tool for dataset curators and researchers.


- Strong Empirical Performance: In the task of detecting known biases, ConceptScope achieves state-of-the-art results, outperforming several baseline methods on benchmarks like NICO++ without requiring the training of additional classifiers specifically for bias detection.

- Automation and Scalability: The framework seems to be designed to be fully automatic, identifying concepts and their distributions without needing costly, fine-grained human annotations for biases.

Weaknesses

- Clarity and Presentation: The overall presentation of the methodology is confusing. Key definitions are either vague or circular. For example, the paper defines "visual concepts" as "groups of visual concepts" and introduces "concept strength" as a critical measure without defining it upfront, making it difficult to understand the logic before reaching later sections. Figure 2, which is central to the method, lacks sufficient detail, particularly regarding the role of the VLM in building the concept dictionary and the degree of manual intervention required.


- Oversimplified Bias Definition: The framework designates a single "bias concept" per class based on the contextual concept with the highest activation strength. This is a restrictive definition, as a class may have multiple spurious correlations (e.g., the "dog" class could be biased toward "grass," "parks," and "tennis balls" simultaneously).


- Potential for Introduced Bias: The initial step of categorizing concepts into "target" and "contextual" may itself introduce bias. For instance, the paper suggests a "smile" is an essential target concept for a "happy face". This assumption could cause the model to incorrectly penalize or misinterpret valid images of happy faces without a visible smile, thereby introducing a new layer of characterization bias.


- Evaluation Gaps: While the method outperforms baselines, the absolute performance on concept segmentation (AUPRC of 0.344) and concept prediction (average AUPRC of 0.66) suggests that the underlying concept extraction has significant room for improvement before it can be considered "accurate" or "reliable" in a standalone capacity. Furthermore, the submission lacks a comparison of the computational cost against baselines, which is an important practical consideration.

---

> ### Author Rebuttal · Authors · 2025-07-31
>
> ***Thank you for the constructive feedback. We hope our responses address the reviewer's concerns.***
>
> ### *Summary of response*
>
> - (W1, Q1, W2, Q2) We **clarify the definition of concept strength**, the **thresholding procedure** for separating context and bias concepts, and the rationale behind it.
> - (W1, Q3) We emphasize that VLM-based annotation is **fully automated** and does **not influence the core algorithm**.
> - (W3, Q4) We demonstrate that alignment scores are **robust across CLIP variants**, supported by controlled experiments with high inter-model agreement.
> - (W4, Q4.3) We address **computational cost**, highlighting a key strength: ConceptScope is **reusable across tasks without retraining**, unlike baselines.
> - (Q4.1, Q4.2) We provide the **missing experimental details**.
>
> ---
> ### *(W1, Q1, W2, Q2) Clarification on concept strength*
> **Concept strength**
>
> We assess non-target concepts using **concept strength** (line 200), defined as the average SAE activation across images of the target class. This continuous metric captures both **frequency** and **magnitude** of concept activation, without relying on arbitrary thresholds.
>
> Due to the sparse nature of activations—typically near zero when absent and high when present—concept strength effectively reflects concept presence. Concepts with significantly high concept strength are labeled as **bias**; others as **context**. To validate its reliability as a frequency proxy, we compared it to a binary frequency measure (activation > 0.5) and found strong agreement (Spearman ρ = **0.73**).
>
> **Concept strength -- Threshold**
>
> We observe that the distribution of concept strength includes a few outliers, suggesting certain concepts are significantly more frequent or strongly activated than others.
>
> To identify these outliers, we apply a threshold of **mean + α × standard deviation** across all context concepts for a given class, with α set to 1 (Eq. 5). This aligns with the **z-score** method—a standard technique for outlier detection [2].
>
> To validate this choice, we compute silhouette scores [1] for varying α values. The score peaks at α = 1, averaging **0.67**, indicating that this setting provides the most coherent separation between **bias** and **context** concepts. Results are averaged over six datasets: Waterbirds, CelebA, Nico++, ImageNet, Food101, and SUN397.
>
> ||α = 0|α = 1|α = 2|α = 3
> |-|-|-|-|-|
> |**AVG**|0.60 (±0.04)|**0.67 (±0.07)**|0.49 (±0.12)|0.16 (±0.25)|
>
> **ConceptScope identifies multiple biases**
>
> This threshold yields an average of **2.4** bias concepts per class. For example, in **Supplementary Fig. 3**, the "freight car" class includes bias concepts like “graffiti” and “rail”. This is intentional—our goal is to identify dominant context concepts likely to reflect dataset artifacts that can mislead model behavior. Empirically, this approach performs well on bias discovery benchmarks (**Tab. 2**) and reveals previously undocumented biases (**Supplementary Fig. 8**), validating its effectiveness.
>
>
> [1] Rousseeuw, P.J. Silhouettes: a graphical aid to the interpretation and validation of cluster analysis. J. Comput. Appl. Math. (1987).
>
> [2] Abdi, H. Z-scores. Encyclopedia of measurement and statistics 3 (2007)
>
> ---
> ### *(W1, Q3) ConceptScope is fully automated*
> The entire ConceptScope pipeline is **fully automated**. Given a class-labeled dataset, it outputs visual concepts with class-dependent categorization with no manual labeling or human feedback.
>
> **Step 1 (Fig. 2(a–b))**: After SAE training, we extract top-activating image patches and corresponding segmentation masks for each latent. To aid interpretation, we use GPT-4o to generate brief descriptions based on visual evidence. The process requires **no manual intervention**, except for minimal prompt tuning—we tried a few prompts and kept the ones that gave qualitatively good results (detailed in **Supplementary Sec. A.2** (lines 52–56) with examples in **Supplementary Fig. 1**). Importantly, this annotation step does **not directly influence** the concept categorization.
>
>
> **Step 2 (Fig. 2(c–d))**: We compute **alignment scores** to separate target vs. context concepts, and use **concept strength** to identify **bias concepts** among the context ones. This step is fully automated and VLM-free.
>
> ---
> ### *(W3, Q4.4) Robustness and validity of target vs. context*
> **Clarification on target and context concept definition**
>
> Let $\mathcal{C} =$ { $c_1, \cdots, c_n$ } be the set of concepts that fully reconstruct an image. For a given concept $c_i$, we compute the alignment score (Eq.(4)) by comparing the model's prediction confidence for the target class when the image is reconstructed using $\mathcal{C} \setminus$ { $c_i$ } (the necessity score in Eq.(3)) or using only { $c_i$ } (the sufficiency score in Eq.(3)), relative to the original image (i.e., reconstructed using $\mathcal C$).
>
> If $\mathcal{C} \setminus$ { $c_i$ } explains the class $y$ significantly *less* than $\mathcal{C}$, while { $c_i$ } alone explains $y$ nearly as well as $\mathcal{C}$, the alignment score is high—indicating that $c_i$ is an important, class-defining **target concept**.
>
> Conversely, if $\mathcal{C} \setminus$ { $c_i$ }explains $y$ almost as well as $\mathcal{C}$, and { $c_i$ } alone fails to explain $y$, the alignment score is low—suggesting that $c_i$ is a non-essential **context concept**.
>
> **Subjectivity of target concept definition**
>
> Given the ambiguity of what constitutes an "essential" concept, there is no gold-standard annotation for labeling concepts as “target” or “context.” This makes evaluation using standard metrics like precision or recall infeasible. Instead, **we conduct a bias discovery experiment** on *known* biases and show that the identified bias concepts are at least as accurate, as shown in **Tab. 2**.
>
> ---
>
> **Robustness to potential bias in CLIP**
>
> CLIP similarity scores [1] are widely used in practice, including for data filtering in LAION-400M [2]. While CLIP may carry pretraining biases, our method is **robust to such biases** by designing the alignment score to depend on the *relative change* in similarity when concepts are ablated or isolated, rather than on absolute similarity values.
>
> **Toy experiment**
>
> To test robustness against potential CLIP bias, we conduct a toy experiment using the Waterbirds dataset [44]. The concept "ocean" is a potential bias for the class "albatross" (a seabird), as they often co-occur. We create two groups of 100 "albatross" images by segmenting the birds and compositing them onto either "ocean" or "forest" backgrounds.
>
> CLIP similarity scores with the text prompt "albatross" are **0.281** (ocean) and **0.252** (forest), slightly favoring ocean. If alignment scores were sensitive to such bias, **ocean** might be misidentified as a target.
>
> However, the alignment scores for **seabird**, **ocean**, and **forest** show that **ocean** and **forest** are consistently categorized as **context concepts**:
> ||seabird|ocean|forest|
> |-|-|-|-|
> |**ocean background**|**1.05**|0.83|0.85|
> |**forest background**|**1.15**|0.88|0.84|
>
> For reference, the average alignment score across datasets is **1.01 ± 0.05** for target concepts and **0.87 ± 0.04** for context, supporting the correct classification in this example.
>
> ---
>
> **Consistency across CLIP variants**
>
> To assess **robustness** to CLIP’s internal biases, we repeat our analysis using three CLIP models trained on different datasets:
> (1) OpenAI CLIP (main), (2) LAION-2B, and (3) DataComp.
>
> We evaluate consistency in concept categorization across Food101, SUN397, and ImageNet. Using Cohen’s kappa [3], we observe an average pairwise score of **0.875**—indicating _almost perfect agreement_—and a standard deviation of alignment scores of **0.110**, showing high consistency despite model differences.
>
> [1] Hessel et al. Clipscore: A reference-free evaluation metric for image captioning. arXiv (2021).
>
> [2] Schuhmann et al. Laion-400m: Open dataset of clip-filtered 400 million image-text pairs. arXiv (2021).
>
> [3] Landis et al. The measurement of observer agreement for categorical data. Biometrics. 1977.
>
> ---
> ### *(W4) Performance on concept segmentation and prediction*
>
> While concept segmentation and prediction can be improved, ConceptScope already outperforms SOTA baselines. Our primary goal is not perfect attribute prediction, but extracting open-vocabulary concepts for **dataset bias analysis**—a capability missing in existing methods.
>
> As shown in **Tab. 2**, **Fig. 4** (main), and **Fig. 8** (supplement), we uncovers *previously unrecognized, non-trivial biases* through concept-level analysis, even without perfect accuracy.
>
> To our knowledge, no prior work systematically evaluates both concept prediction and spatial attribution of SAE as done in Sec. 4.1 and 4.2, marking this as a key contribution of our paper.
>
> ---
> ### *(W4, Q4.3) Computational cost*
> We report the SAE training cost in **Supplementary Section A.1 (lines 37–39)**: ~2,208 GFLOPs per batch (batch size 64) with precomputed CLIP activations. Direct comparison with Tab. 2 baselines is difficult due to differing goals and setups.
>
> ConceptScope performs **a one-time SAE training** on a general dataset (e.g., ImageNet) and can be reused across downstream tasks **without retraining**. In contrast, baselines detect bias via model errors and must **train a new classifier for each dataset**. Although individual classifiers (e.g., ResNet50 at 4 GFLOPs/image) are cheaper, their cost adds up, making baselines less scalable for multi-dataset analysis.
>
> ---
> ### *(Q4.1, Q4.2) Experiment details*
> **Error bars in segmentation task**: The error bars in Fig. 3 represent the standard deviation across 150 classes in the segmentation task.
>
> **Precision@10 metric of bias discovery task**: The details of the bias discovery experiment are described in **Section C** of the Supplementary (lines 208-212).
>
> ### *Typo*
> Thanks for pointing out, we will revise accordingly.

---

> ### Author Response · Authors · 2025-08-07
>
> Dear reviewer, thanks for following up and for your valuable suggestions so far. Below, we clarify your remaining question regarding the concept strength and the number of bias concepts. We also share a summary of paper revisions in a separate comment.
>
> ### Q1: Concept strength
>
> To address your question, we reworked the paragraph "Identifying bias concepts from contextual concepts" (lines 197--205) as follows:
>
> > **Distinguishing bias and context concepts using concept strength.** We leverage the SAE encoder's (non-negative) output $f(\mathbf{z})$ (Eq. 1). For each concept, we average this quantity across all samples $\mathbf{z} \in Z_y$ belonging to class $y$. We then compute the *concept strength*  $\tilde f _{c,y}=\text{avg} _{\mathbf{z} \in Z_y}(f(\mathbf{z}) _c)$ for each for concept $c$. A high concept strength indicates that the concept is frequently and confidently associated with the class $y$.
>
> > Excluding the target concepts $T_y$ of the class, we compute summary statistics $\mu _{y}=\text{avg} _{c'}(\tilde f _{c',y})$ and $\sigma _{y}=\text{std} _{c'}(\tilde f _{c',y})$ for all context concepts $c'\in C \setminus T_y$. We call a concept $c'$ a *bias concept* if its strength exceeds the mean concept strength by more than one standard deviation ($\tilde f _{c',y} \geq  \mu _{y} + \sigma _{y}$) and call it a non-bias context concept otherwise. See Fig. 2 (d) for an illustrative description.
>
> We think this edit contributes to the clarity of the paper.
>
>
>
> ### Q2: Will some concepts always be identified as bias?
>
> Thank you for following up on this. Since we are splitting the non-target concepts based on mean and standard deviation into context and bias, you are right that we will always identify some concepts as bias concepts -- this is by design. Empirically, using our current definition, we find an average of 2.4 ± 0.92 bias concepts per class in ImageNet. All 1,000 classes have at least one concept identifed as a bias concept.
>
> As we noted in our rebuttal ("Concept strength -- Threshold") we could consider different thresholds (0, 2, 3 standard deviations) which would result in a decrease of this number and a more conservative labeling as "bias". The motivation for using one standard deviation was empirically validated by the fact that it well separates the "bias" and "context" concepts as indicated by the silhuette score (see table in rebuttal).
>
> With this definition, we are able to post-hoc analyse the relation of target, context and bias concept strengths across datasets. The following table shows the mean concept strength across **biased** datasets:
>
> | Mean concept strength, biased data |	Waterbird	|CelebA|	Nico++ 	|avg |
> |-|-|-|-|-|
> |target concepts |	0.186 |	0.384 |	0.210 |	 0.260 |
> |context concepts |	0.165 |	0.261 |	0.402 |	0.280 |
> |bias concepts |	0.296 |	0.446 |	0.461 |	0.400 |
>
> In contrast, we see that on ImageNet and Food101 (where less bias is expected), the target concepts always exhibit a higher mean concept strength than context and bias:
>
> | Mean concept strength, "natural" data |	ImageNet	| Food101	| avg |
> |-|-|-|-|
> |target concepts |	0.722 |	0.705	| 0.713 |
> |context concepts |	0.186 |	0.306	| 0.246 |
> |bias concepts | 	0.38 |	0.617	| 0.498 |
>
> We think this is another good validation of our current rationale. We hope this clarifies your question, but we would be happy to discuss further.

---

> ### Author Response · Authors · 2025-08-07
>
> ### Revision plan
>
> > Moreover, can you please indicate how all the points in your rebuttal will be reflected in the manuscript?
>
> In summary, we have already or will revise relevant sections to the rebuttal points:
> - Section 3 (Method): `(W1, Q1, W2, Q2)`, `(W1,Q3)`, `(W3, Q4.4)`
> - Section 4 (SAE experiments): `(W1,Q3)`, `(Q4.1)`
> - Section 5 (ConceptScope experiments): `(W1, Q1, W2, Q2)`, `(W4, Q4.3)`, `(Q4.2)`
> - Section 6 (Conclusion): `(W4)`
>
> Here we provide a point-by-point plan of the revision:
>
> **(W1, Q1, W2, Q2) Clarification on concept strength**
> - Please see our edit to the paper above.
> - We will also add statistics showing multiple biases are captured in **Sec. 5 - Dataset and Model Bias Analysis**.
>
> **(W1, Q3) ConceptScope is fully automated**
> - We will revise **Sec. 3.2 - Interpreting learned concepts** (line 156-166) to include the annotation process detailed in the supplementary material (A.2 line 40-56), and clarify that it does not affect the concept categorization.
> - We will also provide qualitative examples of concept annotations with corresponding images, and consistency analysis across different VLMs at the end of **Sec. 4 - SAEs are Reliable Concept Extractors** with more detail in **Supp. A.2**.
>
> **(W3, Q4.4) Robustness and validity of target vs. context**
> - We will revise **Sec. 3.1 - Problem formulation** (line 112-123) to more clearly define concepts.
> - We will also emphasize that the alignment score is robust to biases in CLIP, supported by toy experiments and consistency across CLIP variants in **Sec. 3.3**, with detail provided in **Supp. A.3**.
>
>
> **(W4) Performance on concept segmentation and prediction**
> - In **Sec. 6 – Conclusion and Limitations** (line 337-342), we will acknowledge that the current concept prediction and segmentation of SAE has room for improvement
> - At the same time, we will highlight ConceptScope's ability to reveal novel biases beyond prior work, even at its current performance level (line 343-349).
>
>
> **(W4, Q4.3) Computational cost**
> - We will explicitly note that the computational costs for training SAE are provided in **Supp. A.1** (line 23-25) and clarify in **Sec. 5.1 - Detecting known biases** that a key advantage of ConceptScope is its reusability across tasks without retraining, unlike existing baselines (supplementary line 251-258).
>
>
> **(Q4.1, Q4.2) Experiment details**
> - We will include all missing experimental details in the main text and guide readers to the relevant supplementary sections.
>
>
> Thanks again for your valuable input, We think it substantially improved the manuscript.

---

> > ### Comment · Reviewer_QZaG · 2025-08-08
> >
> > Dear authors,
> >
> > Thank you for providing the extra clarifications and the plan for how you are going to incorporate the changes in the manuscript. The rebuttal has addressed my concerns and I raised my score accordingly.
> >
> > Kind regards

---

> > > ### Author Response · Authors · 2025-08-08
> > >
> > > Dear Reviewer, thank you for your reply. We greatly appreciate the constructive discussion and your positive feedback.

---

### Official Review · Reviewer_P7Ht · 2025-07-02

**Clarity:** 2
**Significance:** 2
**Originality:** 3
**Rating:** 4
**Confidence:** 3

**Summary:**

The paper proposes ConceptScore, a framework for discovering bias concepts in class-labeled image datasets. It first uses SAE and VLM to build a concept dictionary from a pretrained model and an image dataset. Then it calculates several metrics based on SAE activations to determine the set of "target", "bias", "non-bias contextual" concepts for a given class y. Experiments demonstrate SAE is able to accurately capture and localize concepts, and on bias discovery task (both known and "in the wild") it outperforms existing methods.

**Questions:**

See weaknesses.

**Ethical Concerns:**

["NO or VERY MINOR ethics concerns only"]

**Final Justification:**

Concerns were mostly addressed in rebuttal, and the remaining ones were understandable.

**Limitations:**

Yes

**Paper Formatting Concerns:**

No.

**Quality:**

3

**Strengths And Weaknesses:**

Strengths:
1. The tackled problem is very important and underaddressed, especially evidenced by recent research.
2. The framework of conception dictionaries together with target, bias, and non-bias contextual concepts is novel and intuitive.
3. The usage of SAE is well motivated and demonstrated to be effective.
4. The experiments conducted are diverse, covering both the SAE part, and the whole pipeline, with different kinds of datasets and tasks.

Weaknesses:

1. A core contribution is the definition to distinguish the concepts into 3 categories. However, I find this part not clear, and it is a very important part to improve.

The difference between the alignment scores (equation (4)) and the bias score (equation (5)) should be clarified. Are they highly correlated? What is the difference intuitively? To the readers they all seem to rank the relevance/co-occurance of the concepts in images of class y. In figure 2 (c) confirm this impression where the target concept also ranks higher in formula (5) but its definition is from eqn (4). The scores should be explained in an intuitive manner, not just mathematical languages, especially in clarifying the differences when it is quite subtle.

Also, the threshold in both (4) and (5) require some explanation/motivation.



2. The necessity and sufficient score computation uses CLIP embedding of class label y as the "ground truth" - masked images with retained patches that generate closer image embeddings to this y embedding are considered closer to the "true" class definition. This could be problematic as CLIP itself could be biased in *its* training. The CLIP text encoder may already have learned to associate beaches with turtle, thus when the image contains beach but not turtle, CLIP can still give a high score. This would disturb the calculation of the scores and make it an inaccurate estimation.



3. The final and very important step of identifying the concept dictionary is using a VLM (e.g., GPT-4o) to annotate, given a few concept-masked referenced images. This step is only described in one sentence. It would be great if more details are given, e.g., the prompt used, how confident the answers are, and how well it aligns with human judgement. Some examples can be presented.

4. This work tackles image datasets with classes (e.g., imagenet), but image-text datasets are gaining dominance, especially in MLLM and generative model training. The work would be more significant if it tackles image-text datasets. This is also acknowledged by authors in the limitation section.

Minor:

Line 202-203: above the average of that of other context concept. Should it be "above one standard deviation to the right of the average..."?

---

> ### Author Rebuttal · Authors · 2025-07-31
>
> ***Thank you for the constructive feedback. We hope our responses address the reviewer's concerns.***
>
> ### *Summary of responses*
>
> - (W1) We **clarify concept categorization** with both intuitive and mathematical explanations, and **validate threshold**s using silhouette scores.
> - (W2) We show that alignment scores are **robust across CLIP variants** through controlled experiments with high inter-model agreement.
> - (W3) We emphasize that VLM-based annotation is **fully automated** and does not affect the core algorithm.
> - (W4) We demonstrate **broader applicability** by evaluating on the MSCOCO image captioning dataset.
>
> ---
> ### *(W1) Clarification on concept categorization*
> **Alignment score**
>
> The **alignment score** quantifies how representative and essential a concept is for predicting a specific class $y$, with higher values indicating greater importance. Let $\mathcal{C} =$ { $c_1, \cdots, c_n$ } be the set of concepts that fully reconstruct an image. For a given concept $c_i$, we compute the alignment score by comparing the model's prediction confidence for the target class when the image is reconstructed using $\mathcal{C} \setminus $ { $c_i$ } (the necessity score in Eq.(3)) or using only { $c_i$ } (the sufficiency score in Eq.(3)), relative to the original image (i.e., reconstructed using $\mathcal C$).
>
> If $\mathcal{C} \setminus$ { $c_i$ } explains the class $y$ significantly *less* than $\mathcal{C}$, while { $c_i$ } alone explains $y$ nearly as well as $\mathcal{C}$, the alignment score is high—indicating that $c_i$ is an important, class-defining **target concept**.
>
> Conversely, if $\mathcal{C} \setminus$ { $c_i$ } explains $y$ almost as well as $\mathcal{C}$, and { $c_i$ } alone fails to explain $y$, the alignment score is low—suggesting that $c_i$ is a non-essential **context concept**.
>
> **Alignment score -- Threshold**
>
> We observe that per-class alignment score distributions typically form **two clusters**—high and low—corresponding to **target** and **context** concepts. To separate them, we apply a threshold of **mean + α × standard deviation**.
>
> We evaluate this separation using the **silhouette score** [1], which measures clustering quality. The score peaks near α = 0, supporting the use of the mean (i.e., normalized alignment score > 0; line 194) as a robust, generalizable threshold. Results are averaged over six datasets: Waterbirds, CelebA, Nico++, ImageNet, Food101, and SUN397.
>
> | |  α = -2 | α = -1 | α = 0| α = 1 | α = 2 |
> | - | - | - | - | - | - |
> | **AVG** | 0.05 (±0.10) |0.27 (±0.15)|**0.59 (±0.08)**|0.54 (±0.10)|0.32 (±0.20)|
>
> ---
>
> **Concept strength**
>
> We assess non-target concepts using **concept strength** (line 200), defined as the average SAE activation across all images labeled with the target class. This continuous metric captures both the **frequency** and **magnitude** of concept activation, without relying on arbitrary thresholds.
>
> Due to the sparse nature of activations—typically near zero when absent and high when present—concept strength effectively reflects concept presence. Concepts with significantly high strength are labeled as **bias concepts**; others as **context concepts**.
>
> To validate its reliability as a frequency proxy, we compared it to a binary frequency measure (activation > 0.5) and found strong agreement (Spearman ρ = **0.73**).
>
> **Concept strength -- Threshold**
>
> We observe that the distribution of concept strength includes a few outliers, suggesting certain concepts are significantly more frequent or strongly activated than others.
>
> To identify these outliers, we apply a threshold of **mean + α × standard deviation** across all context concepts for a given class, with α set to 1 (Eq. 5). This aligns with the **z-score** method—a standard technique for outlier detection [2].
>
> To validate this choice, we compute silhouette scores [1] for varying α values. The score peaks at α = 1, averaging **0.67**, indicating that this setting provides the most coherent separation between **bias** and **context** concepts. Results are averaged over six datasets: Waterbirds, CelebA, Nico++, ImageNet, Food101, and SUN397.
>
> | | α = 0 | α = 1| α = 2 | α = 3 |
> | - | - | - | - | - |
> |**AVG**|0.60 (±0.04)|**0.67 (±0.07)**|0.49 (±0.12)|0.16 (±0.25)|
>
>
> [1] Rousseeuw, P.J. "Silhouettes: a graphical aid to the interpretation and validation of cluster analysis." Journal of computational and applied mathematics (1987).
>
> [2] Abdi, H. "Z-scores." Encyclopedia of measurement and statistics 3 (2007)
>
> ---
> ### *(W2) Robustness to potential bias in CLIP*
> CLIP similarity scores [1] are widely used in practice, including for data filtering of LAION-400M [2]. Although CLIP may carry biases from its pretraining, our method is **robust to such potential biases** by designing the alignment score to rely on the *relative change* in similarity when specific concepts are ablated or isolated in the image, rather than on absolute similarity values.
>
> **Toy experiment**
>
> To test robustness against potential CLIP bias, we conduct a toy experiment using the Waterbirds dataset [44]. The concept "ocean" is a potential bias for the class "albatross" (a seabird), as they often co-occur. We create two groups of 100 "albatross" images by segmenting the birds and compositing them onto either "ocean" or "forest" backgrounds.
>
> CLIP similarity scores with the text prompt "albatross" are **0.281** (ocean) and **0.252** (forest), slightly favoring ocean. If alignment scores were sensitive to such bias, **ocean** might be misidentified as a target.
>
> However, the alignment scores for **seabird**, **ocean**, and **forest** show that **ocean** and **forest** are consistently categorized as **context concepts**:
>
> |                      | seabird | ocean | forest |
> |----------------------|---------|-------|--------|
> | **ocean background** | **1.05**| 0.83  | 0.85   |
> | **forest background**| **1.15**| 0.88  | 0.84   |
>
> For reference, the average alignment score across datasets is **1.01 ± 0.05** for target concepts and **0.87 ± 0.04** for context, supporting the correct classification in this example.
>
> ---
>
> **Consistency across CLIP variants**
>
> To assess **robustness** to CLIP’s internal biases, we repeat our analysis using three CLIP models trained on different datasets:
> (1) OpenAI CLIP (main), (2) LAION-2B, and (3) DataComp.
>
> We evaluate consistency in concept categorization across Food101, SUN397, and ImageNet. Using Cohen’s kappa [3], we observe an average pairwise score of **0.875**—indicating _almost perfect agreement_—and a standard deviation of alignment scores of **0.110**, showing high consistency despite model differences.
>
> [1] Hessel, J. et al. "Clipscore: A reference-free evaluation metric for image captioning." arXiv (2021).
>
> [2] Schuhmann et al. (2021). Laion-400m: Open dataset of clip-filtered 400 million image-text pairs. arXiv (2021).
>
> [3] Landis JR, Koch GG. The measurement of observer agreement for categorical data. Biometrics. 1977.
>
> ---
> ### *(W3) Explanation of VLM-based concept annotation*
> **Purpose and role of VLM annotation**
>
> Using VLM for concept annotation is a post-hoc step intended to make the extracted concepts easier for human interpretation. Importantly, this step does **not directly influence** the concept categorization, which includes alignment score or concept strength computation and thresholding.
>
>
> **Detailed annotation procedure**
>
> The full procedure, including the prompt template, is described in **Supplementary Section A.2**, with examples in **Supplementary Fig. 1**. We follow PatchSAE [28] to collect top-activating image patches and their segmentation masks, then use GPT-4o to generate concise natural language descriptions. The process is fully automated, aside from light prompt tuning to select a format that consistently produces accurate and meaningful labels.
>
>
> **Annotation quality**
>
> Although we do not manually verify every annotation, qualitative inspection shows that the generated descriptions are consistent and meaningful across similar visual patterns (see **Supplementary Fig. 1** and **Fig. 4**). Examples include concise, interpretable phrases like "knitted fabric" or "yellow objects."
>
> To assess consistency, we compared generated captions by GPT-4o and LLaVA-NeXT (`llama3-llava-next-8b`). The average cosine similarity was **0.754** (±0.119) in CLIP’s text embedding space and **0.862** (±0.050) in OpenAI’s `text-embedding-ada-002` space, indicating strong agreement and convergence toward simple, intuitive descriptions. Examples of caption pairs:
> - LLaVA-NeXT vs. GPT-4o generated -- CLIP similarity, OpenAI similarity
> - curtains vs. curtains -- 1.00, 1.00
> - rooster vs. red rooster -- 0.81, 0.93
> - vehicles vs. convertible cars -- 0.76, 0.88
> - textured vs. knitted fringe -- 0.53, 0.81
>
> We acknowledge that annotation quality may vary across different VLMs and consider alignment with human perception an important area for future work.
>
> ---
> ### *(W4) Generalization to image-text dataset*
>
> To demonstrate generality beyond single-label classification, we apply our method to a **multi-label** setting using the MSCOCO 2017 test set [1]. We extract object mentions from ground-truth captions as multi-label targets and conduct concept categorization as in **Section 5.2** (lines 305–311). Representative findings include:
> - **Cat**: Bias – “indoor environments”; Context – “office workstation”, “bed sheets”.
> - **Dog**: Bias – “couch”, “living room”; Context – “grass”.
> - **Bird**: Bias – “bird feeder”, “trees”; Context – “cage”, “blue sky”.
>
> These results show that our method effectively captures concept-level biases and contexts in complex, real-world multi-label scenarios.
>
> [1] Lin, T. et al. "Microsoft coco: Common objects in context." ECCV 2014.
>
> ---
> ### *Typo*
> Thanks for pointing out, we will revise accordingly.

---

> > ### Author Response · Authors · 2025-08-05
> > **Gentle reminder for discussion**
> >
> > Dear Reviewer,
> >
> > Thanks again for your thoughtful review. We addressed your questions and would appreciate if you could re-evaluate our contribution in the light of our rebuttal. If concerns are remaining, we are happy to discuss further during the discussion period.

---

> > ### Comment · Reviewer_P7Ht · 2025-08-05
> >
> > Thanks to the authors for the rebuttal. Most issues addressed, or understandable (e.g., the bias introduced by using CLIP). I hope the authors can continue to improve the clarity of the key definitions, ideally with examples illustrating their differences, in the final revision. I think the problem this paper addresses is important, and I will change my rating to borderline accept.

---

> ### Author Response · Authors · 2025-08-06
>
> We sincerely appreciate your positive assessment of our rebuttal. As suggested, we will revise the manuscript to improve the clarity of key definitions. If you have any further questions or require additional clarification, please let us know.

---

### Author Response · Authors · 2025-08-09
**Summary of Responses to Reviews and Revision Plan (1 / 2)**

Dear AC and reviewers,

We want to thank all reviewers and the AC again for their time and careful feedback. Based on the comments, we made substantial improvements over the submitted manuscript and provided detailed plans for revision.

Overall, Reviewers `P7Ht`, `6bLm`, and `QZaG` found our rebuttal convincing and explicitly raised their scores. We have provided additional explanations to address some remaining concerns for Reviewers `usgc`.

Building on the strengths that the reviewers already highlighted—novelty (`P7Ht`, `QZaG`, `6bLm`, `usgc`), practical importance (`P7Ht`, `QZaG`, `6bLm`), and strong empirical results (`P7Ht`, `QZaG`, `usgc`)—we have:
1. improved readability and clarity of method description,
2. experimentally validated design choices,
3. provided missing details for completeness, and
4. demonstrated the generalizability and broader applicability of the suggested framework.

---

### 1. Improved readability and clarification of the method description

We have revised `Sec. 3.3 lines 173-196` of the manuscript as follows:
> **Separating target and context concepts.** To distinguish **target** and **context** concepts for a class $y$, we define a *class--concept alignment score* that measures how representative and essential a concept is for predicting $y$.
Let $\mathcal{C} =$ { $c_1, \ldots, c_M\$} be the set of learned concepts. For a given concept $c$, we compare the model’s prediction confidence in three cases: (1) with all concepts $\mathcal{C}$, (2) without $c$ ($\mathcal{C} \setminus ${ $c$ }), and (3) with only $c$ ({$c$}).
The set $\mathcal{C} \setminus $ { $c$ } is obtained by masking out the spatial attribution of $c$ ($m_c(x)$) from the image $x$ ($x \odot (1 - m_c(x))$), while {$c$} uses the segmentation mask directly ($x \odot m_c(x)$).

> We formalize comparison with two metrics: *necessity* $N(c,y)$ and *sufficiency* $S(c,y)$.
Necessity measures the drop in similarity when the concept region is removed (i.e., case (1) vs. (2)); sufficiency measures the similarity when only the concept region is retained (i.e., case (1) vs. (3)):$$
\small
\mathrm{N}(c, y) = \frac{1}{|X_y|} \sum_{x \in X_y}
\frac{P(y \mid x)}{P\bigl(y \mid x \odot (1 - m_c(x)) \bigr)},
\quad
\mathrm{S}(c, y) = \frac{1}{|X_y|} \sum_{x \in X_y}
\frac{P\bigl(y  \mid x \odot m_c(x)\bigr)}{P(y \mid x)},
$$ where $x$ is an image of class $y$, $X_y$ is the set of such images, $P(y \mid x)$ denotes the model's prediction confidence for class $y$ given $x$, measured by a cosine similarity between the CLIP text embedding of $y$ and the image embedding of $x$, $m_c(x)$ is the binary mask for concept $c$, and $\odot$ is element-wise multiplication (Fig.2 (c)). We combine the two metrics into an alignment score $A(c,y) = \frac{\mathrm{N}(c,y) + \mathrm{S}(c,y)}{2}$. A concept is labeled **target** if $A(c,y) \ge \mu^{\text{align}}_y$ (class mean), and **context** otherwise. In Appendix A.3, we provide details and empirical demonstration of the proposed method, along with illustrative examples in Appendix Fig. 2.

`Sec. 3.3 lines 197-205` as follows:
> **Distinguishing bias and context concepts using concept strength.** We leverage the SAE encoder's (non-negative) output $f(\mathbf{z})$ (Eq. 1). For each concept, we average this quantity across all samples $\mathbf{z} \in Z_y$ belonging to class $y$. We then compute the *concept strength*  $\tilde f_{c,y}=\text{avg}_{\mathbf{z} \in Z_y}(f(\mathbf{z})_c)$ for each for concept $c$. A high concept strength indicates that the concept is frequently and confidently associated with the class $y$.

> Excluding the target concepts $\mathcal{T} _y$ of the class, we compute summary statistics $\mu^{\text{c.s.}} _{y}=\text{avg} _{c'}(\tilde f _{c',y})$ and $\sigma^{\text{c.s.}} _{y}=\text{std} _{c'}(\tilde f _{c',y})$ for all context concepts $c'\in \mathcal{C} \setminus \mathcal{T} _y$. We call a concept $c$ a *bias concept* if its strength exceeds the mean concept strength by more than one standard deviation ($\tilde f _{c,y} \geq  \mu^{\text{c.s.}} _{y} + \sigma^{\text{c.s.}} _{y}$) and call it a non-bias context concept otherwise. See Fig. 2 (d) for an illustrative description.

Additionally,
- In `Sec. 3.2 lines 156-166`,  we include the annotation process originally detailed in the `Supp. A.2 line 40-56`  and clarify that the process is fully automated and does not affect the concept categorization.
- In `Sec. 5.3 lines 322-335`, we revise the explanation of how we simulate concept-based distribution shifts and assess model robustness without requiring a separate out-of-distribution dataset.

---

> ### Author Response · Authors · 2025-08-09
> **Summary of Responses to Reviews and Revision Plan (2 / 2)**
>
> ### 2. Experimentally validated design choices
>
> - In `Supp. A.3 line 65` we add **A.3.1 Robustness of alignment score to potential bias in CLIP** where we demonstrate the robustness of the alignment scores to internal biases in CLIP through toy experiments involving background changes and an analysis of score consistency across different CLIP variants.
> - We also add **A.3.2 Empirical validation of thresholds** to provide empirical validation for the thresholds used to distinguish target vs. context concepts and context vs. bias concepts.
>
> ---
>
> ### 3. Provided missing details for completeness
> - In `Supp. A.2 line 40` we add **Generated annotation quality** with examples of concept annotations and an analysis of their consistency across different VLMs.
> - In `Sec. 2 lines 77–85`, we include additional related work on bias identification and clearly distinguish our approach, noting that, unlike prior work focused on model bias, ConceptScope directly analyzes the training dataset.
> - In `Sec. 5.1 lines 296-304`  and `Sec. 5.2 lines 306-311`, we add statistics showing that multiple biases are captured and report the average number of activated concepts per dataset.
> - We include all previously missing experimental details in the main text and guide readers to the relevant supplementary sections, and add the remaining rebuttal content to the supplementary material.
>
> ---
>
> ### 4. Demonstrated generalizability and broader applicability
> - In `Sec. 6` we add **Discussion** on how ConceptScope can be applied to domains significantly different from ImageNet and on the coverage of discovered concepts.
> - Additionally, in `Sec. 6`, we include a **Broader impact** note showing that our method can also be applied to image–text caption datasets.
> - We also add a `Limitations (line 337-342)` noting that although concept prediction and segmentation performance can be further improved, ConceptScope already reveals novel biases beyond prior work at its current performance level.
>
> ---
>
> **Thank you again for the time invested in reviewing our work.**

---

### Decision · Program_Chairs · 2025-09-17

**Decision:**

Accept (poster)

**Comment:**

This paper proposes a method to characterize dataset biases by concepts it discovers. Instead of relying on predefined attributes or manual labels, the paper explains present biases in a dataset by concepts automatically detected. The reviewers found the paper was well written and the evaluation was thorough.

The paper also shows that the method can detect "novel" biases (not from any existing labels or predefined categories) in a dataset and the reviewers thought this would be practically very useful. The exact same idea and general pipeline have already been published in a previous paper [1], however this paper was not cited. The AC conditionally recommends acceptance of the paper, provided that the authors include missing references in discussion and clarify the novelty of the current paper in comparison to these papers. It is also strongly recommended that the authors perform a comparative evaluation between them.

[1] Yang, Y., Kim, S., & Joo, J. (2022). Explaining deep convolutional neural networks via latent visual-semantic filter attention. In Proceedings of the IEEE/CVF Conference on Computer Vision and Pattern Recognition (pp. 8333-8343).